# Large-Scale Targeted Cause Discovery via Learning from Simulated Data

**Jang-Hyun Kim**[*]                                                                                   *janghyun@mllab.snu.ac.kr*
*Department of Computer Science, Seoul National University*

**Claudia Skok Gibbs**                                                                                 *csg337@nyu.edu*
*Center for Data Science, New York University*

**Sangdoo Yun**                                                                                        *sangdoo.yun@navercorp.com*
*NAVER AI Lab*

**Hyun Oh Song**                                                                                       *hyunoh@mllab.snu.ac.kr*
*Department of Computer Science, Seoul National University*

**Kyunghyun Cho**                                                                                      *kyunghyun.cho@nyu.edu*
*Center for Data Science, New York University*
*Prescient Design, Genetech*

**Reviewed on OpenReview:** *https://openreview.net/forum?id=NVgy29IQw8*

## Abstract

We propose a novel machine learning approach for inferring causal variables of a target variable from observations. Our focus is on directly inferring a set of causal factors without requiring full causal graph reconstruction, which is computationally challenging in large-scale systems. The identified causal set consists of all potential regulators of the target variable under experimental settings, enabling efficient regulation through intervention. To achieve this, we train a neural network using supervised learning on simulated data to infer causality. By employing a subsampled-ensemble inference strategy, our approach scales with linear complexity in the number of variables, efficiently scaling up to thousands of variables. Empirical results demonstrate superior performance in identifying causal relationships within large-scale gene regulatory networks, outperforming existing methods that emphasize full-graph discovery. We validate our model's generalization capability across out-of-distribution graph structures and generating mechanisms, including gene regulatory networks of E. coli and the human K562 cell line. Implementation codes are available at https://github.com/snu-mllab/Targeted-Cause-Discovery.

## 1 Introduction

Identifying causal relationships among variables is a fundamental challenge in machine learning, with widespread applications in generative modeling, system explanation, and variable control (Pearl, 2009). Conventional methods have attempted to infer causal structures using statistical tests or by fitting probabilistic models to observations (Spirtes et al., 2001; Chickering, 2002). However, the exponential complexity of causal graphs renders the problem NP-hard (Chickering, 1996), posing significant challenges for large-scale systems (Zanga et al., 2022).

In large-scale systems, inferring the complete set of causal relationships is often unnecessary, as various downstream tasks directly benefit from identifying only the causes that are relevant to specific outcomes

---

[*]Work done while visiting New York University.

(Magliacane et al., 2016). For instance, in drug discovery, it is sufficient to identify causal transcription factors associated with disease-related genes, rather than mapping the entire gene regulatory network (GRN), to effectively guide therapeutic interventions (Huynh-Thu et al., 2010). Similarly, in epidemiology or policy evaluation, interventions can be effectively designed by pinpointing only the key causal drivers of infection rates or targeted outcomes, without modeling the full complexity of inter-variable dependencies (Glass et al., 2013). These examples illustrate that identifying the causes of specific targets has practical utility while reducing the complexity inherent in discovering entire causal structures.

In this study, we propose a scalable method for identifying the causal variables of a target variable from observations in large-scale systems, by leveraging data simulators. Our approach aims to identify both direct and indirect causes that can regulate the target variable under experimental conditions (Figure 1). Rather than reconstructing an entire causal graph, we focus on identifying a set of causal factors, simplifying the estimation process while retaining practical utility. We introduce this problem setting as **targeted cause discovery**.

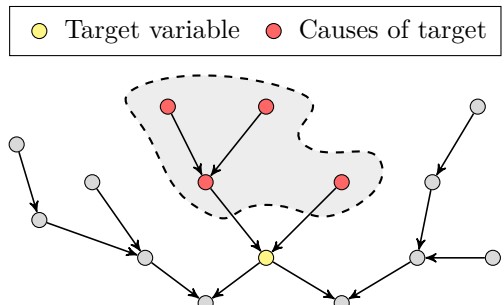

Figure 1: Illustration of targeted cause discovery in a causal graph. Instead of inferring the full causal graph structure, we identify a set of causal variables for a target.

An alternative to our proposed method involves inferring causal variables by traversing the ancestors of a target within a causal graph derived from existing causal discovery techniques. However, estimating complete causal graphs becomes computationally prohibitive for systems with thousands of variables (Zanga et al., 2022). Furthermore, imperfect inference from limited observations leads to error accumulation at each step of ancestral traversal, resulting in exponentially increasing inaccuracies. We explore this issue in greater detail in Section 3.

To overcome these limitations, we propose an end-to-end machine learning approach that directly estimates a set of causal variables from observational and experimental data. Our method trains a deep neural network on simulated data to learn a causal discovery algorithm that generalizes to unseen causal structures (Ke et al., 2023). This data-driven approach allows the model to implicitly capture assumptions embedded in the simulated data. A key advantage of our method is to learn complex causal mechanisms from data, such as differential equations, without requiring explicit formal modeling in score-based causal discovery methods (Chickering, 2002; Zheng et al., 2018). We employ the Transformer architecture for our causal discovery model (Vaswani et al., 2017), building on its demonstrated effectiveness in previous works (Lorch et al., 2022). Nonetheless, the quadratic complexity of Transformer's attention mechanism presents computational challenges. To mitigate this, we introduce a subsampled-ensemble inference strategy that scales linearly with the number of variables, enabling efficient application to large-scale problems involving thousands of variables.

Empirical evaluations demonstrate that our method effectively identifies causal relationships within complex systems involving thousands of variables, under the availability of high-fidelity simulators (Section 5). We demonstrate our model's capability to generalize from random causal structures to real-world GRNs, including E. coli, yeast, and the K562 human cell line (Dibaeinia & Sinha, 2020). We further assess the robustness and generalization capability of our approach through comprehensive evaluations on both synthetic and real-world datasets, examining a range of graph structures, causal mechanisms, and levels of simulator fidelity, thereby underscoring its applicability in practical scenarios.

## 2 Preliminary and Related Work

### 2.1 Problem Formulation

We consider a set of random variables $\mathcal{V} = \{x_1, \ldots, x_n\}$ that has a causal structure represented by a directed acyclic graph (DAG), $\mathcal{G} = (\mathcal{V}, \mathcal{E})$. We assume the data-generating process involves no latent variables, and for simplicity, we assume there is no selection bias (Pearl, 2009). The joint distribution $p(\mathcal{V})$ is defined

as $\prod_i p(x_i \mid \text{Pa}(x_i))$ where $\text{Pa}(x_i)$ means the set of parent variables of $x_i$ in $\mathcal{G}$. We obtain observations of variables through ancestral sampling.

**Definition 1.** *Causes for a target node $x_i$ is defined as the set of ancestors of $x_i$ in $\mathcal{G}$, denoted by $\text{An}(x_i)$.*

**Definition 2.** *A variable $x_j$ is a marginal cause of $x_i$ if and only if $\exists c$ s.t.*

$$p(x_i \mid \text{do}(x_j = c)) \neq p(x_i).$$

The operator $\text{do}(\cdot)$ represents an intervention that fixes specific variables to predefined values during the ancestral sampling process (Pearl, 2009). Specifically, the observation for $\mathcal{V}$ under $\text{do}(x_j = c)$ follows $\delta_c(x_j) \prod_{i \neq j} p(x_i \mid \text{Pa}(x_i))$, where $\delta_c$ is a Dirac delta function at $c$. Definition 2 implies that interventions on causal variables make a distributional change in the target variable.

**Proposition 1.** *The set of marginal causes of $x_i$ is a subset of $\text{An}(x_i)$.*

We provide the proof for Proposition 1 in Appendix A. This proposition implies that the ancestor set $\text{An}(x_i)$ forms a necessary condition for identifying variables that influence the target variable $x_i$ in experimental settings. Specifically, in large, sparse networks such as GRNs, efficiently identifying these influential variables can facilitate effective control of target variables. We refer to the task of identifying the set of causes $\text{An}(x_i)$ of a target variable $x_i$ as *targeted cause discovery*.

## 2.2 Amortized Variational Inference for Causal Structure Discovery

Classical causal discovery aims to infer the causal structure $\mathcal{G}$ from observation. Lorch et al. (2022) introduced a Bayesian perspective on causal discovery by defining a joint distribution over causal structures and mechanisms as $p(\mathcal{G}, \phi)$, where $\phi$ represents mechanism and noise parameters. Given this prior distribution, the likelihood of observed variables $\mathcal{V}$ can be expressed as $p(\mathcal{V} \mid \mathcal{G}, \phi)$. Marginalizing out the mechanism parameters $\phi$ yields the posterior distribution $p(\mathcal{G} \mid \mathcal{V})$, which is typically intractable. Learning-based causal discovery approaches tackle this intractability by approximating the posterior using amortized variational inference parameterized as $q_\theta(\mathcal{G} \mid \mathcal{V})$ (Lorch et al., 2022; Ke et al., 2023).

Rather than estimating the entire graph structure, we specifically focus on inferring the ancestor set $\text{An}(x_i)$ of a target variable $x_i$, represented by the posterior $p(\text{An}(x_i) \mid \mathcal{V})$. This ancestor distribution is inherently determined by the posterior over the entire graph $p(\mathcal{G} \mid \mathcal{V})$. One possible method is to first fit $q_\theta(\mathcal{G} \mid \mathcal{V})$ and then approximate $p(\text{An}(x_i) \mid \mathcal{V})$ through graph sampling, resembling approximate Bayesian computation (Sisson et al., 2018). However, in large-scale scenarios, such sampling-based methods based on the graph posterior distribution become computationally expensive and present additional technical challenges, as detailed in Section 3. In our approach, we directly infer the posterior distribution over the ancestor set by optimizing variational parameters $\theta$ of a variational distribution $q_\theta(\text{An}(x_i) \mid \mathcal{V})$.

## 2.3 Related Work

**Causal discovery** Conventional causal discovery methods aim to infer the exact causal graph structure $\mathcal{G}$ from observations (Spirtes et al., 2001; Chickering, 2002). These approaches can be broadly categorized into constraint-based and score-based methods. Constraint-based methods use conditional independence tests and formalized directional decision rules to identify causal relationships (Spirtes et al., 1995; 2001). However, they require independence testing over combinatorial sets of variables, leading to exponential complexity in the number of variables (Colombo et al., 2014). Subsequent works have sought to reduce this complexity by leveraging sparsity assumptions in causal structures with Gaussian mechanisms (Kalisch & Bühlman, 2007) or by improving computational efficiency through parallelization (Le et al., 2016). Score-based methods, on the other hand, optimize the goodness-of-fit of graph structures while balancing complexity constraints (Chickering, 2002; Hauser & Bühlmann, 2012; Zheng et al., 2018; Brouillard et al., 2020). To navigate the combinatorial search space of causal graphs, these approaches rely on specific assumptions about graph structures and data-generating mechanisms (Lopez et al., 2022). An alternative line of research focuses on estimating the topological ordering of causal variables, which helps reduce optimization complexity (Reisach et al., 2021; Sanchez et al., 2023).

**Local approaches** To address the computational challenges of estimating full causal graphs, some efforts have focused on inferring the local causal structure of a target node, such as the Markov blanket (Aliferis et al., 2010; Wu et al., 2019), parent set (Gao & Ji, 2015), or parent-cause set (Yin et al., 2008). However, these approaches still suffer from an exponential search space, requiring efficient algorithms for large-scale settings (Gao & Ji, 2015). Magliacane et al. (2016) introduced the problem of estimating ancestral structures. The key difference from our work is that their approach considers the ordering of ancestral nodes (or causes), whereas we do not impose any ordering constraints. By removing the order constraint, we reformulate our training objective as a binary classification task—determining whether a given node is a cause of the target based on observations of all other nodes. This simplification enables us to develop a scalable method for both training and inference.

**Learning-based approaches** Recent efforts have introduced learning-based approaches that leverage large datasets and computational power for causal discovery (Lopez-Paz et al., 2015; Löwe et al., 2022; Lorch et al., 2022; Ke et al., 2023; Wu et al., 2024). These methods generate synthetic data with known ground-truth causal graphs and train neural networks to infer graph structures from observations (Lorch et al., 2022). In this work, we analyze the generalization performance of these data-driven approaches in large-scale, complex systems. Specifically, we propose a novel strategy that identifies a set of causal variables for a target with linear complexity, enabling efficient scaling to thousands of variables.

## 3 Targeted Cause Discovery versus Causal Structure Discovery

Targeted cause discovery offers several technical advantages over causal structure discovery, particularly in addressing the shortcomings of conventional methods when the primary objective is to identify the causes of a specific target. In this section, we compare our approach, which directly estimates the set of causes, with an alternative method that infers causes by traversing the ancestors of the target within an estimated causal graph.

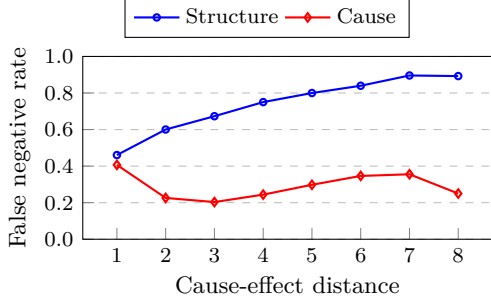

Figure 2: Targeted cause discovery error rate as a function of shortest path length between nodes. *Cause* refers to our method, directly estimating the causes of a target. *Structure* denotes a method that infers causes from an estimated causal graph. Both methods use the same model architecture and dataset but differ in training objectives and inference strategies. Appendix B.6 provides detailed experimental setting.

1) Error propagation in estimation: With a limited number of observations, causal structure discovery methods have prediction errors (Lorch et al., 2022). When causes are inferred from an inaccurately estimated causal graph, these errors propagate exponentially: If a method has an expected prediction error rate of $e$ per edge, the error rate for estimating causes at a distance $d$ from the target is approximately $1 - (1 - e)^d$. Our approach mitigates this issue by directly inferring causality among distant variables, avoiding the error propagation inherent in graph-based traversal. Figure 2 presents empirical measurements of cause prediction error rates across varying cause-effect distances, where we compare methods estimating the full causal structure with those identifying only a set of causes. To ensure a fair comparison, we use the same model architecture and dataset for both methods, differing only in training objectives and inference strategies. The results indicate that the error rate of a method identifying causes remains stable regardless of distance, whereas the error rate of a method trained to estimate the causal structure increases as the cause-effect distance grows.

2) Technical challenges in training: In learning-based causal discovery, neural networks are typically trained to predict the adjacency matrix of a causal structure (Ke et al., 2023;

Table 1: Averaged ratio of causes (or parents) per node, *i.e.*, $|\text{An}(x)|/n$ (or $|\text{Pa}(x)|/n$). We draw statistics from 10 graphs, each with 1000 nodes and an average in-degree of 2.

| Graph type | Parent | Cause |
|---|---|---|
| Erdős–Rényi | 0.2% | 1.1∼1.2% |
| Scale-free | 0.2% | 0.4∼2.2% |
| Gene regulatory | 0.2% | 0.5∼1.1% |

Lorch et al., 2022). However, due to the sparsity of causal graphs, the resulting binary adjacency matrix is highly imbalanced, posing significant training challenges. For instance, in the E. coli GRN, there are only about 2.3 edges per node among 1,565 nodes (Dibaeinia & Sinha, 2020). Such an extreme imbalance is known to hinder neural network training by providing sparse learning signals from the loss function (Kaur et al., 2019). In contrast, our approach trains neural networks to predict ancestors rather than the full adjacency matrix. This formulation mitigates the sparsity issue as shown in Table 1, thereby improving training stability and effectiveness.

3) Subsampled-inference guarantee: Targeted cause discovery enables subsampled inference, allowing the identification of causal relationships using only a subset of the system's variables (Proposition 2). Specifically, a cause of a target variable $x_i$ remains a cause within a subsampled variable system (Bongers et al., 2016). This property is particularly useful in large-scale settings, where processing data from all variables is computationally infeasible. Leveraging this property, we propose an efficient algorithm capable of scaling to thousands of variables, as detailed in Section 4.2. It is worth noting that subsampled inference is non-trivial for causal structure discovery, as the direct causal structure can change within a subsampled variable system. We provide a proof of this proposition in Appendix A.

**Proposition 2.** *For any variable subset $V \subset \mathcal{V}$ s.t. containing $x_i$ and all root variables in $\mathcal{G}$, let $\text{An}(x_i; V)$ denote the set of ancestors of $x_i$ in the system consisting of $V$, following the marginalization rule of Bongers et al. (2016). Similarly, let $\text{Pa}(x_i; V)$ denote the set of parents of $x_i$ within $V$. Then, $\text{An}(x_i; V) = \text{An}(x_i) \cap V$. However, a counterexample exists for parents: $\text{Pa}(x_i; V) \neq \text{Pa}(x_i) \cap V$.*

## 4 Method

In this section, we present our method for Targeted Cause Discovery with Data-driven Learning, termed **TCD-DL**. We consider a system with $n$ variables $\mathcal{V} = \{x_1, \ldots, x_n\}$ having an underlying causal structure $\mathcal{G}$. We denote the observation data as $X \in \mathbb{R}^{n \times m}$, where $m$ is the number of observation samples. In the interventional setting, we define a boolean matrix $M \in \{0, 1\}^{n \times m}$, where 1 indicates the occurrence of interventions.

Our objective is as follows: Given a target variable $x_i \in \mathcal{V}$, predict a label $l_i \in \{0, 1\}^n$ from the observation $X$ and intervention matrix $M$, where $l_i[j] = 1$ indicates that $x_j$ is a cause of $x_i$, *i.e.*, $x_j \in \text{An}(x_i)$. We adopt a binary labeling approach, disregarding proximity in the causal graph, as quantifying causal contributions based on proximity is non-trivial. To tackle this problem probabilistically, we estimate a continuous **cause score** vector $s_i \in \mathbb{R}^n$, where $s_i[j]$ represents the unnormalized likelihood of $x_j$ being a cause of $x_i$. A higher score corresponds to a greater likelihood of causality. This continuous relaxation enables the use of gradient-based optimization techniques, which are well-suited for large-scale settings (Bottou, 2010). To obtain a set of estimated causes, we apply thresholding to $s_i$, considering values greater than zero as causes.

### 4.1 Data-Driven Learning

A straightforward approach to discovering causality involves conducting interventions on every single variable with a statistically sufficient number of trials to confirm the hypothesis. However, it is often impractical to intervene at every single variable due to experimental limitations and the high costs associated with conducting a sufficient number of trials (Addanki et al., 2020).

To address this challenge, we develop a parameterized model $f_\theta$ capable of inferring causality among variables from observations $X \in \mathbb{R}^{n \times m}$ of arbitrarily limited size. Given the target variable index $i$, the model processes the entire dataset $X$ with intervention matrix $M$ and returns a cause score vector $s_i \in \mathbb{R}^n$ as

$$s_i = f_\theta(X, M, i). \tag{inference}$$

By leveraging observations from all nodes, we can infer causal relationships more accurately than relying solely on the observations of a single pair of variables. We discuss this in more detail in Section 5.4.

We implement $f_\theta$ as a deep neural network and train on simulated data $\mathcal{D} = \{(X_k, M_k, \mathcal{G}_k) \mid k \in \mathcal{I}\}$ using a supervised learning approach. Here, $\mathcal{I}$ denotes an index set, and $X_k$ represents an observation dataset

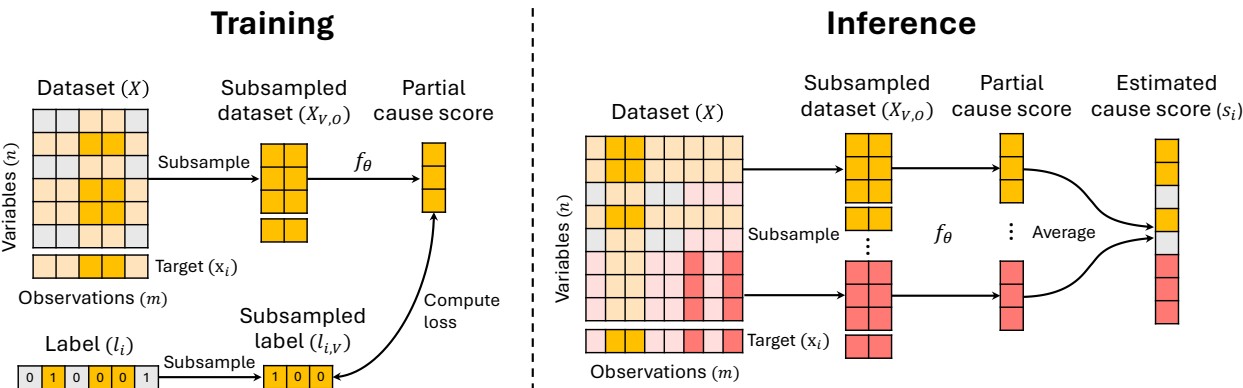

Figure 3: **Overview of our method.** The left figure depicts a single training step, while the right figure illustrates the inference procedure with multiple subsampling and ensembling. Note that the intervention matrix $M$ undergoes the same subsampling as $X$, resulting in the stacked input $[X_{V,O}, M_{V,O}]$ of shape $n' \times m' \times 2$, which is then fed into the model $f_\theta$. We omit the intervention matrix in the figure for simplicity.

sampled from a synthetic causal graph $\mathcal{G}_k$ with an intervention matrix $M_k$. We generate synthetic datasets using a simulator on random graphs, as detailed in Section 5. Empirically, we demonstrate that the model generalizes to unseen causal structures and causal mechanisms, achieving strong performance on real-world datasets (Section 5.3). For a causal graph $\mathcal{G}_k$ with $n_k$ variables, we obtain a label $l_{k,i} \in \{0,1\}^{n_k}$ for each variable $i = 1, \ldots, n_k$ using topological sorting. The label is defined such that $l_{k,i}[j] = 1$ if the $j$-th variable is a cause of the $i$-th variable in $\mathcal{G}_k$. Our training objective, with prediction $f_\theta(X_k, M_k, i) \in \mathbb{R}^{n_k}$, is

$$\underset{\theta}{\text{minimize}} \ \mathbb{E}_{k\sim\mathcal{I}}\mathbb{E}_{i\sim\{1,\ldots,n_k\}}[\mathcal{L}(f_\theta(X_k, M_k, i), l_{k,i})], \qquad \text{(training)}$$

where $\mathcal{L}$ is the loss function. In this study, we use binary cross-entropy with logits for $\mathcal{L}$ (Wei et al., 2022), ignoring the loss at the target node's position.

**Acyclicity.** Our problem focuses on estimating a set of causes rather than inferring the full causal graph structure, and therefore does not require explicitly enforcing an acyclicity constraint. Moreover, our supervised learning approach leverages training labels extracted from a DAG, which contains no cycles. Training on such data provides implicit guidance for our model to generate causal predictions that are consistent and non-contradictory.

**Model architecture.** The model $f_\theta$ comprises two sequential modules, a feature extractor $g_\theta$ and a score calculator $h$, designed to optimize compute efficiency. The feature extractor $g_\theta$ processes the stack $[X, M] \in \mathbb{R}^{n \times m \times 2}$ to produce features $F_1, F_2 \in \mathbb{R}^{n \times d}$, where each variable index $i$ corresponds to two $d$-dimensional features $F_1[i], F_2[i] \in \mathbb{R}^d$. We employ an axial-Transformer as the feature extractor (Ho et al., 2019), a matrix-input variant of Transformers that has been a conventional architecture in prior learning-based approaches (Lorch et al., 2022; Ke et al., 2023). Notably, our framework is compatible with general model architectures designed for matrix-shaped inputs. A detailed description of the model architecture is provided in Appendix B.1.

The score calculator $h$ computes the cause score vector $s_i$ for a target index $i$ using dot-product between features as $h(F_1, F_2, i) = F_1 F_2[i] \in \mathbb{R}^n$, where $F_1 \in \mathbb{R}^{n \times d}$ and $F_2[i] \in \mathbb{R}^d$. To sum up, $f_\theta(X, M, i) = h(g_\theta(X, M), i) = h(F_1, F_2, i)$. The use of a separate feature extractor allows for the reuse of features across multiple target indices $i$, enhancing training efficiency with batch data processing.

### 4.2 Subsampled Inference for Scaling Up

The axial Transformer comprises three main operations: two attention layers over each variable and observation dimension, and a feed-forward layer (Ho et al., 2019). For an input $X \in \mathbb{R}^{n \times m}$, the complexities of

attention layers are $O(n^2m)$ and $O(nm^2)$, while the complexity of the feed-forward layer is $O(nm)$. The quadratic complexity of the attention mechanism poses challenges for large inputs in terms of computational time and memory usage. In Appendix B.4, we present a detailed analysis of the experimental scales and discuss the computational challenges.

To address this issue, we propose a subsampled inference strategy (Figure 3), supported by Proposition 2, which estimates the causes of a target variable using subsampled variables and observations. Specifically, we define subsample sizes $n' \ll n$ for variables and $m' \ll m$ for observations, set independently of $n$ and $m$, based on available computing resources. Given a target variable $\mathrm{x}_i$, we randomly sample a subset of variables $V \subset \mathcal{V}$ with $|V| = n'$, ensuring $\mathrm{x}_i \in V$. We then extract the corresponding observation matrix $X_V \in \mathbb{R}^{n' \times m}$ and intervention matrix $M_V \in \{0,1\}^{n' \times m}$. To avoid distortion from interventions on variables outside the subsampled set $V$, we select only those observations (columns of $X_V$) where no variables in $\mathcal{V} \setminus V$ are intervened. From these, we randomly subsample $m'$ observations, yielding $X_{V,O} \in \mathbb{R}^{n' \times m'}$ and $M_{V,O} \in \{0,1\}^{n' \times m'}$, where $O$ denotes the selected observation indices. We denote this subsampling process as $V, O \sim S(X, M, i)$.

We then infer the causal effects on $\mathrm{x}_i$ from variables in $V$ using $f_\theta(X_{V,O}, M_{V,O}, i) \in \mathbb{R}^{n'}$. Repeating this process across multiple subsamplings of variables and observations, we aggregate and average the results to compute the final cause score vector $s_i$. We call this process subsampled-ensemble inference. In practice, we perform subsampling to ensure that each variable is considered identically $T$ times (Algorithm 2). For a variable $\mathrm{x}_j \in V$, we denote the estimated cause score to $\mathrm{x}_i$ as $f_\theta(X_{V,O}, M_{V,O}, i)[j]$. (Here, $j$ refers to the index in $\mathcal{V}$, but for clarity, we retain the same index notation within $V$.) The final estimation is then

$$s_i[j] = \mathbb{E}_{V,O \sim S(X,M,i)} \left[ f_\theta(X_{V,O}, M_{V,O}, i)[j] \mid \mathrm{x}_j \in V \right]. \qquad \text{(subsampled-ensemble inference)}$$

For training, we apply the identical random subsampling strategy on inputs and target labels. For $X_k$ and a target variable index $i$, we denote the subsampled data as $X_{k,V,O} \in \mathbb{R}^{n' \times m'}$ and the corresponding target label as $l_{k,i,V} \in \{0,1\}^{n'}$. In practice, we generalize the results from Proposition 2, defining $\mathrm{An}(\mathrm{x}_i; V) = \mathrm{An}(\mathrm{x}_i) \cap V$ for arbitrary subsets of variables. Specifically, each label entry is 1 if the corresponding variable belongs to $\mathrm{An}(\mathrm{x}_i) \cap V$, and 0 otherwise. This ensures consistent labeling, $i.e.$, for $\mathrm{x}_j \in \mathrm{An}(\mathrm{x}_i)$, the variable $\mathrm{x}_j$ receives a label of 1 regardless of $V$, providing consistent supervision signals throughout training. The subsampled version of our training objective becomes

$$\underset{\theta}{\text{minimize}} \ \mathbb{E}_{k \sim \mathcal{I}} \mathbb{E}_{i \sim \{1,...,n_k\}} \mathbb{E}_{V,O \sim S(X_k, M_k, i)}[\mathcal{L}(f_\theta(X_{k,V,O}, M_{k,V,O}, i), l_{k,i,V})]. \qquad \text{(subsampled training)}$$

We optimize $f_\theta$ using stochastic gradient descent with the AdamW optimizer (Loshchilov & Hutter, 2019). Algorithms 1 and 2 describe pseudo codes of our final training and inference algorithms. We leave detailed hyperparameters in Appendix B.2.

**Algorithm complexity.** Our algorithm reduces the inference complexity from quadratic to linear with respect to the number of variables $n$. Specifically, we fix the subsample size $n'$ as a constant across all experiments, regardless of the variable size $n$. By splitting the inputs into fixed-size chunks of $n'$ and repeating the Transformer computation $n/n'$ times, we achieve a computational complexity of $O(n'^2 \cdot n/n') = O(nn')$, reducing the original complexity of $O(n^2)$. Proposition 3 provides a detailed complexity analysis of our inference algorithm, with a proof included in Appendix A.

**Proposition 3.** *Let $n'$ and $m'$ denote the subsampled variable and observation sizes, and let $T$ denote the ensemble size. For a dataset $X$ with $n$ variables, the inference complexity of our algorithm is $O(nm'T(n' + m'))$.*

**Algorithm 1** Training (batch version)

> **inputs**: $\mathcal{D} = \{(X_k, M_k, \mathcal{G}_k) \mid k \in \mathcal{I}\}$
> **parameters**: subsample sizes $n'$ and $m'$, batch size $b$
> initialize $\theta$
> **repeat**
>  $\mathcal{X}, \mathcal{Y} \leftarrow \emptyset, \emptyset$
>  **for** $j = 1$ to $b$ **do**
>   $k \leftarrow$ sample from $\mathcal{I}$
>   $V, O \leftarrow$ subsample given $X_k, M_k$
>   $\mathcal{X} \leftarrow \mathcal{X} \cup \{(X_{k,V,O}, M_{k,V,O}, i) \mid \mathrm{x}_i \in V\}$
>   $\mathcal{Y} \leftarrow \mathcal{Y} \cup \{l_{k,i,V} \mid \mathrm{x}_i \in V\}$
>  **end for**
>  $\mathbf{g} \leftarrow$ calculate gradients $\nabla_\theta \mathcal{L}(f_\theta(\cdot), \cdot)$ on $\mathcal{X}, \mathcal{Y}$
>  $\theta \leftarrow$ update using gradients $\mathbf{g}$
> **until** convergence
> **return** $\theta$

**Algorithm 2** Inference

> **inputs**: $X \in \mathbb{R}^{n \times m}$, $M \in \{0,1\}^{n \times m}$, target index $i$
> **parameters**: subsample sizes $n'$ and $m'$, #ensemble $T$
> initialize $s_i \leftarrow \mathbf{0}_n$
> **for** $t = 1$ to $T$ **do**
>  $I \leftarrow \mathrm{permute}(\{1, \ldots, n\} \setminus \{i\})$
>  split $I = \cup_{j=1}^{b} I_j$ where $b = \lceil \frac{n}{n'} \rceil$ and $|I_j| \leq n'$
>  **for** $j = 1$ to $b$ **do**
>   $I_j \leftarrow I_j \cup \{i\}$ and $V \leftarrow \{\mathrm{x}_k \mid k \in I_j\}$
>   $O \leftarrow$ subsample given $V, X, M$
>   $s_i[I_j] \leftarrow s_i[I_j] + f_\theta(X_{V,O}, M_{V,O}, i)$
>  **end for**
> **end for**
> $s_i \leftarrow s_i / T$
> **return** $s_i$

## 5 Experiment

We present an experimental analysis of out-of-distribution (OOD) settings for targeted cause discovery. In Section 5.2, we verify that our model, trained on random graphs, effectively identifies causal relationships in biological networks consisting of several thousand variables. In Section 5.3, we examine the model's generalization capabilities with respect to novel causal mechanisms and varying noise levels, including analyses on a real-world human gene dataset. Finally, in Section 5.4, we analyze interventional settings and assess sensitivity to hyperparameters. To compare against conventional algorithms such as PC and GIES, we conduct additional small-scale experiments detailed in Appendix C.2.

### 5.1 Setup

**Simulated dataset.** We generate the training set $\mathcal{D}$ and test set using the SERGIO GRN simulator (Dibaeinia & Sinha, 2020). This simulator produces single-cell gene expression data, modeling the stochastic nature of transcription based on a user-provided causal graph structure. We conduct experiments over varying levels of simulator's observational fidelity, where higher fidelity yields expression data closer to the population parameters. Details about the simulator, including the generation mechanisms and the definition of fidelity levels, are provided in Appendix B.3. For the training dataset, we use random graph structures including Erdős–Rényi (ER), Scale-Free (SF), and Stochastic Block Model (SBM) with 1,000 variables (Drobyshevskiy & Turdakov, 2019). The test data is generated from biological structures of E. coli (1,565 genes) and yeast (4,441 genes) as obtained from Marbach et al. (2009).

**Intervention.** We adopt the intervention settings from Lorch et al. (2022). Specifically, we perform single-variable interventions by knocking out gene expressions, *i.e.*, setting their transcription rates to zero. We generate 10 samples per intervention, along with 500 observational samples (details provided in Appendix B.3). In Figure 8, we analyze model performance under various intervention scenarios, including different ratios of intervened nodes and multi-variable interventions.

**Identifiability.** Our method shares similar identifiability considerations with existing learning-based approaches (Lorch et al., 2022; Ke et al., 2023). In particular, we focus on experimental settings with single-node interventions, where the true causal structure is theoretically identifiable in the limit of infinite interventional data from each variable (Eberhardt et al., 2006). In practice, however, the number of available interventional samples is typically limited, precluding full identifiability guarantees. To address this, we propose a probabilistic framework designed to provide calibrated estimates of causal relationships given the available data, rather than enforcing strict identifiability. This design reflects a practical trade-off, enabling the model to

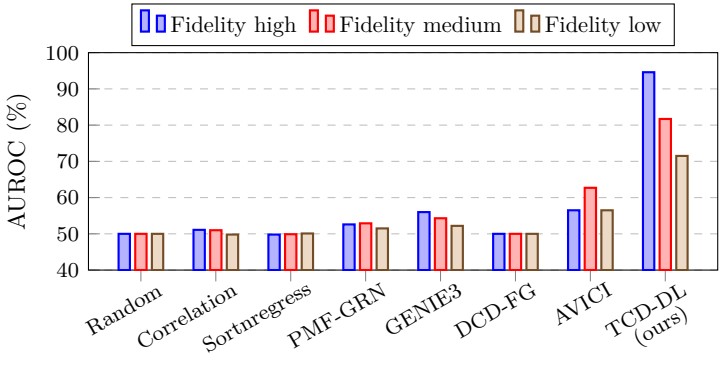 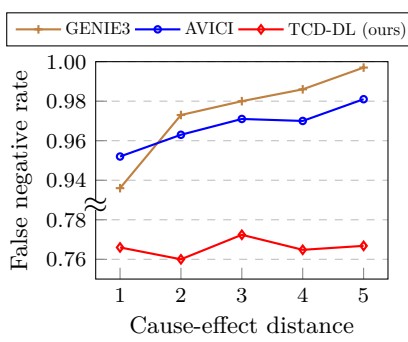

(a) Performance across varying levels of simulator fidelity      (b) Error by causal distance

Figure 4: **Benchmarking results.** (a) Performance on E. coli GRN with 1565 genes over varying levels of simulator's observational fidelity. We provide AUROC, AP, and F1 score values, including standard deviations in Table 10. (b) The cause prediction error rate as a function of the shortest path length between variables in a causal graph.

generate causal predictions that are consistent with observed data, even in low-sample regimes. Empirically, we find that our approach yields meaningful and robust causal inferences under realistic data constraints, as illustrated in Figure 8.

**Baseline.** We comprehensively evaluate causal discovery methods drawn from various methodological frameworks. As baselines, we include a *random* guessing model, the absolute *correlation* score, and a regression-based method known as *sortnregress* (Reisach et al., 2021). Additionally, we compare against the score-based approach DCD-FG (Lopez et al., 2022), an improved method designed to address instability issues by DCDI (Brouillard et al., 2020) and NO-TEARS (Zheng et al., 2018) in large-scale settings (Kaiser & Sipos, 2022). We evaluate the learning-based approach AVICI (Lorch et al., 2022) by utilizing their released model, which was trained on the same simulator and comparable amount of data as ours. It is worth noting that AVICI closely aligns with the concurrent method CSIVA (Ke et al., 2023), while CSIVA does not provide source code or trained models. Conventional algorithms such as PC and GIES (Spirtes et al., 2001; Hauser & Bühlmann, 2012) are evaluated separately in small-scale experiments due to their limited scalability (Appendix C.2). We assess representative GRN inference methods, including the tree-based method GENIE3 (Huynh-Thu et al., 2010) and the linear factor model PMF-GRN (Skok Gibbs et al., 2024). All mentioned methods, except PC, compute likelihood scores indicative of (direct) causal relationships among variables. We utilize these likelihood scores to construct baseline causal score vectors $s_i$.

**Evaluation metric.** We evaluate the targeted cause discovery performance on variables having at least one causal variable. For each target variable, we compare the predicted cause score vector against the ground-truth binary label, where 1 indicates a causal relationship between variables. We employ AUROC, Average Precision (AP), and F1 score for this binary classification task (Rainio et al., 2024). To calculate the F1 score, we threshold the cause scores to match the number of positive predictions with the ground-truth labels. We measure average performance across all valid variables (those with at least one cause) in test datasets. We obtain statistics using expression data with 10 different random seeds for sampling.

## 5.2 Generalization on Unseen Causal Structure

**Benchmarking.** Figure 4-a illustrates the performance of targeted cause discovery on the E. coli GRN, which has a novel causal structure not observed during training. Our approach consistently achieves the best performance by a large margin, demonstrating the effectiveness of our data-driven learning approach. The results reveal the shortcomings of existing methods relying on specific assumptions. Linear models (correlation, sortnregress) fail to identify causality in our settings with complex generation mechanisms. As a sanity check, we observe that correlation achieves 70.7% AUROC on causal graphs with linear generation

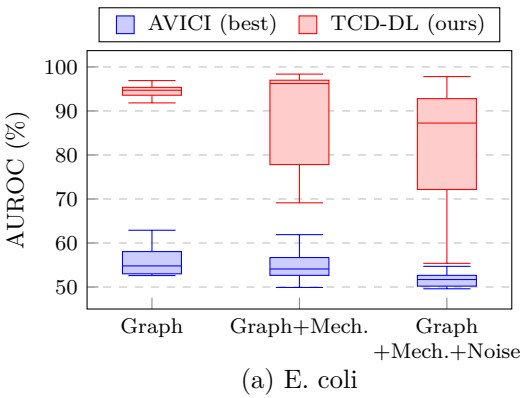
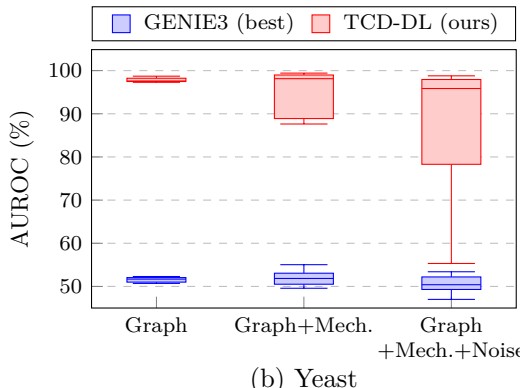

(a) E. coli            (b) Yeast

Figure 5: **OOD performance.** Box plots illustrating targeted cause discovery performance on novel graph structures, mechanism parameters (mech.), and noise configurations unseen during training. For the E. coli dataset, AVICI is the best-performing baseline method, whereas GENIE3 achieves the best results for yeast.

mechanisms, showing moderate performance under valid assumptions. The score-based approach (DCD-FG) performs nearly at random, likely due to invalid assumptions about the factorizability of graph structures and simplistic generation mechanisms (Lopez et al., 2022). Our method outperforms the learning-based approach (AVICI), which focuses on direct causality and does not leverage subsampled inference, thereby demonstrating the effectiveness of our scalable algorithm.

**Robustness to causal distance.** To gain deeper insight into the observed performance improvements, we measure the false-negative rate as a function of the shortest path length in the ground-truth causal graph (Figure 4-b). To ensure a fair comparison, we threshold the causal scores of the best-performing methods so that each produces an identical number of positive predictions. The results indicate that baseline methods exhibit increasing error rates as causal distance grows, whereas our method maintains consistent performance. These findings highlight a key advantage of our approach that reliably identifies both proximal and distant causes without performance degradation.

**Error analysis.** While our model demonstrates strong performance, there still remain errors, particularly as simulator fidelity decreases. To investigate the sources of these errors, we compare our model's performance on random graphs sampled from the training setting (*i.e.*, validation set) to its performance on E. coli graphs (*i.e.*, test set). Table 2 reveals that both validation and test performance decline as simulator fidelity decreases.

Table 2: AUROC (%) on random graphs (validation) and E. coli GRN (test).

| Data \ Fidelity | High | Medium | Low |
|---|---|---|---|
| Validation | 92.6 | 83.3 | 70.1 |
| Test | 94.6 | 81.7 | 71.5 |

This parallel degradation suggests that the primary source of error is not a generalization issue, but rather stems from other factors. This finding raises questions about causal identifiability in low-fidelity scenarios, indicating that the dataset itself may lack sufficient information for accurate causal inference.

**Runtime measurement.** The runtime of our inference algorithm for processing each target variable is 2.5 seconds for E. coli (1,565 genes) and 7.8 seconds for yeast (4,441 genes), as measured with an NVIDIA RTX 3090 GPU. These results highlight that our model, with subsampled-ensemble inference, operates within seconds for large-scale systems. We provide a comparison of the runtime with baseline methods in Appendix C.1, where some baselines, such as DCD-FG and PMF-GRN, take several hours for inference.

### 5.3 Extended Out-of-Distribution Analysis

**OOD simulator configuration.** We further study the generalization capabilities of our model by testing on causal mechanism and noise configurations that differ from the training. We adopt the configurations from Lorch et al. (2022), as described in Appendix B.3. Figure 5 shows that our method largely outperforms the baselines across all settings, demonstrating robust generalization. Notably, when only the graph structure

differs from training, our model shows high prediction capability. However, as generation mechanisms diverge from the training setting, the performance variance increases. These findings show both the strengths of our approach and the challenges inherent in generalizing to diverse generation mechanisms.

**Real-world human cell analysis.** We test our simulator-trained model in a real-world scenario using a Perturb-seq dataset derived from the K562 cell line of a patient with chronic myelogenous leukemia (Replogle et al., 2022). We focus on the gene MYC, a key oncogene involved in cell proliferation, growth, and apoptosis, frequently overexpressed in cancers (Dhanasekaran et al., 2022). We compute cause scores for 1,868 genes and select the top 20 genes as predicted causes of MYC expression (Appendix B.5). One major challenge here is that the exact structure of the human GRN has not yet been fully elucidated. To evaluate our predictions, we compare them with the following three sources: (1) The STRING database, which provides associations between proteins but does not indicate the causal direction (Szklarczyk et al., 2023). (2) Existence of literature that supports causality between a specific gene pair (Table 7). (3) Top 20 genes with the highest expression correlations to MYC. Figure 6 shows that our model demonstrates strong predictive accuracy, achieving 90% precision compared to STRING. Notably, 30% of our predictions are validated by existing literature (5 greens and 1 yellow), demonstrating that our model uncovers meaningful causal relationships. An interesting observation is that only the gene EEF1A1 shows high expression correlation with the target, suggesting that our method identifies novel causal factors not captured by correlation

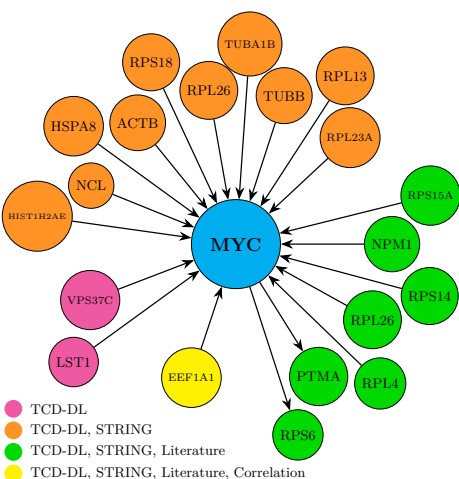

Figure 6: Causal genes of MYC identified by our method. Genes are categorized by validation from STRING, existing literature, and top-20 expression correlations. We note that existing literature validates PTMA and RPS6 as effects of MYC.

ranking. In Appendix C.3, we provide additional analysis, including the quantitative comparison to correlation ranking and the predicted influence of causal genes. These results highlight our model's potential for real-world applications in understanding and manipulating gene regulation, particularly in the context of personalized medicine and targeted cancer therapies.

**Ablating training sources.** To further assess the generalization capabilities of our approach, we perform ablation studies by excluding specific causal structures and mechanism types from the training data, and evaluating the resulting models. In Figure 7-a, we conduct experiments with various graph structures, including Erdős–Rényi (ER), directional Scale-Free (SF-direct), Scale-Free (SF), and the Stochastic Block Model (SBM) (Drobyshevskiy & Turdakov, 2019). In Figure 7-b, we vary the causal generation mechanisms to include different analytic functions (linear, non-linear multi-layer perceptron (MLP), polynomial, and sigmoid), while maintaining a scale-free network structure, following Wu et al. (2024).

Figure 7 illustrates the relative performance for each test case, scaled from 0 (random) to 100 (best). Both subfigures exhibit similar patterns, where diagonal entries have relatively lower performance, indicating a generalization gap when the testing data type is omitted from the training sources. Nonetheless, relative performance consistently exceeds 90, highlighting the robust generalization capability of our method. Notably, models trained on a diverse combination of all data types consistently achieve near-optimal performance. This data-scaling effect is consistent with observations from large-scale language models (Brown et al., 2020), suggesting that diversifying training data improves overall causal discovery performance.

## 5.4 Additional Analysis

**Ratio of intervened nodes.** We evaluate our model by varying the ratio of intervened nodes and the number of intervention samples per node on the E. coli test dataset. To maintain a fixed dataset size, we adjust the number of observational samples accordingly in each scenario. We select intervened nodes randomly, and use a single model for evaluation across all settings. In this experiment, we disable the dropout

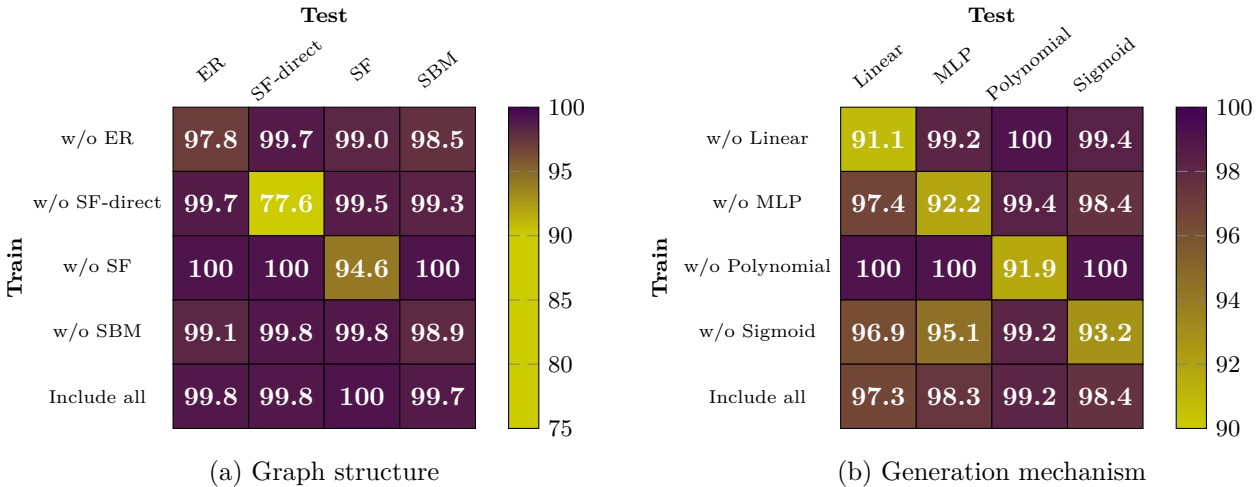

(a) Graph structure                    (b) Generation mechanism

Figure 7: **Training source ablation analysis.** The metric used is the relative AUROC, with 100 indicating the best model's performance on each test case and 0 corresponds to random prediction, *i.e.*, $100(p - p_{\text{random}})/(p_{\text{best}} - p_{\text{random}})$, where $p$ denotes the AUROC score of a given model.

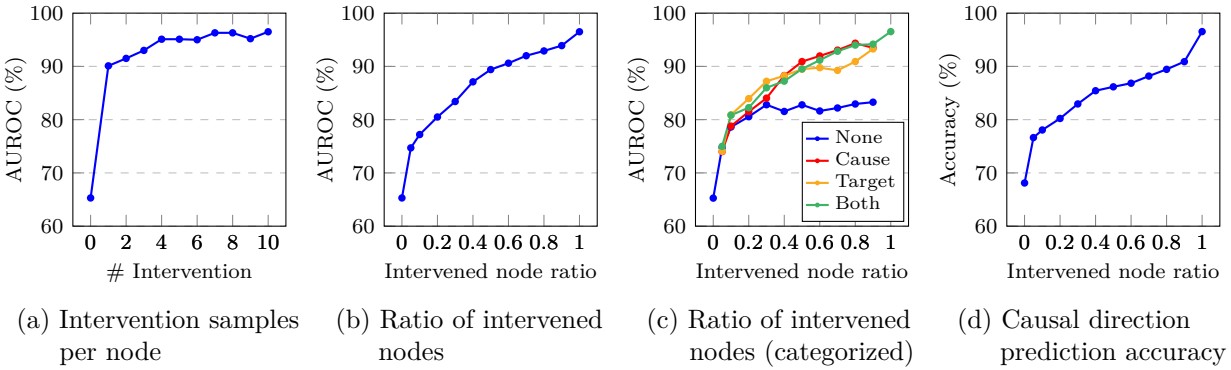

(a) Intervention samples    (b) Ratio of intervened    (c) Ratio of intervened    (d) Causal direction
per node                        nodes                        nodes (categorized)        prediction accuracy

Figure 8: **Intervention analysis.** Performance on the E. coli GRN test dataset with varying intervention settings. (a) Controlling the number of intervention samples per node in the test dataset. (b) Controlling the ratio of intervened nodes among all nodes. A zero ratio indicates that only the observational samples are used. (c) Performance with categorized intervention types. We categorize target nodes into four cases and report average scores for each case: no intervention on the target and ancestors (*None*), intervention on at least one ancestor and not on the target (*Cause*), intervention on the target and not on ancestors (*Target*), and intervention on both (*Both*). (d) Causal direction prediction accuracy of our model across varying ratios of intervened nodes.

effect in the simulator (Appendix B.3), as the random removal of information can weaken interventional signals and lead to misleading analyses.

From Figure 8-a, we observe that even with only one intervention sample per node (complemented by observational data), the model achieves reasonable performance, reaching an AUROC of 90.1%. As we decrease the ratio of intervened nodes (Figure 8-b), performance decreases sublinearly. These results demonstrate that our method effectively leverages interventional information to infer causality. Furthermore, the model consistently provides meaningful predictions, even with limited intervention data, and surprisingly, performs moderately well with purely observational data, achieving 65.3% AUROC. In Figure 8-c, we present categorized performance based on whether the target node itself or one of its causes has been intervened upon, while varying the ratio of intervened nodes. Interestingly, as indicated by the *None* category in the figure,

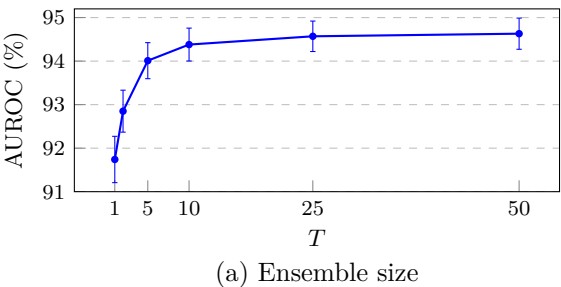
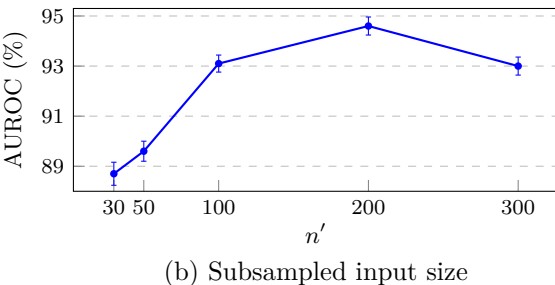

(a) Ensemble size

(b) Subsampled input size

Figure 9: **Hyperparameter analysis.** Performance on E. coli GRN with varying (a) ensemble sizes $T$ and (b) input variable subsample sizes $n'$.

we observe that even when neither the target node nor its causes are intervened on, the model achieves moderate performance ($\sim$82% AUROC) by leveraging interventional information on other variables.

**Multi-node intervention.** We conduct experiments involving multi-node interventions by increasing the number of interventions per sample from 1 to 3 during both training and testing while keeping all other configurations unchanged. In this scenario, we observe a slight performance drop from 94.6% to 93.7% AUROC on the E. coli test dataset. We suspect this decline results from increased complexity due to the combinatorial explosion in possible interventions, making it more challenging for the model to fully capture relevant causal information. Nonetheless, exploring optimal intervention settings remains an important direction for future research.

**Causal direction prediction.** To verify that our model indeed infers causality beyond mere correlation learning, we measure the causal direction prediction accuracy across varying ratios of intervened nodes (Figure 8-d). Specifically, for every pair of nodes $i$ and $j$ with a known causal relationship, we estimated the causal direction by comparing the cause scores $s_i[j]$ and $s_j[i]$ predicted by our model. If $s_i[j] > s_j[i]$, we predicted $j$ as the cause of $i$. Note that random guessing achieves an accuracy of 50% for this task. As shown in Figure 8-d, our model accurately predicts causal directions under the full-node intervention setting, while achieving 68% accuracy in the observational setting. This clearly demonstrates that our model indeed infers causality beyond mere correlation learning.

**Algorithm hyperparameter.** We analyze the impact of design choices in our subsampled-ensemble inference strategy by sweeping the ensemble size $T$ and the subsampled variable size $n'$ in Algorithm 2. Figure 9-a demonstrates that performance improves as ensemble size increases, plateauing around 25. This result validates the effectiveness of our ensembled inference approach. Figure 9-b illustrates the effect of subsampled variable size per input. Performance increases with input size up to 200, suggesting that processing larger variable sets through a single Transformer forward pass allows the model to utilize richer relational information. However, performance declines for input sizes exceeding 300, indicating conflicting effects between input complexity and information richness. These results validate our approach of processing a subset of variables, highlighting its effectiveness compared to processing all variables simultaneously.

## 6 Conclusion and Discussion

In this work, we propose an effective and scalable approach for targeted cause discovery, aiming to identify a set of causal variables for a given target. Our method trains a neural network to learn causal discovery algorithms from simulated data. To handle large-scale systems, we introduce a subsampled-ensemble inference strategy that achieves linear complexity with respect to the number of variables. Empirical results on gene regulatory networks demonstrate that our approach significantly outperforms existing causal discovery baselines, exhibiting strong generalization across diverse graph structures and generation mechanisms. By shifting the focus from explicit causal modeling to a data-driven framework, our method aligns with recent advances in deep learning. We anticipate further performance gains through data scaling within our

scalable framework. However, our reliance on data-driven black-box models comes at the cost of reduced interpretability. Balancing causal interpretability with the advantages of data-driven methods remains a crucial future direction for targeted cause discovery.

## Broader Impact Statement

Our work introduces a scalable technical approach to targeted cause discovery in large-scale causal inference problems, specifically designed for computational efficiency and robustness in identifying relevant causal variables. Positive impacts of our method include improved computational performance and increased accuracy in practical causal inference tasks, enabling researchers to efficiently analyze complex systems such as gene regulatory networks. Although this research primarily advances foundational machine learning techniques without direct societal applications, potential negative impacts might arise if applied irresponsibly—for example, misinterpretation of causal findings leading to incorrect conclusions in biological or medical contexts.

## Acknowledgement

The work was supported by the National Science Foundation (under NSF Award 1922658). Kyunghyun Cho is supported by the Samsung Advanced Institute of Technology (under the project Next Generation Deep Learning: From Pattern Recognition to AI). Jang-Hyun Kim and Hyun Oh Song are supported by SNU-NAVER Hyperscale AI Center, Institute of Information & Communications Technology Planning & Evaluation (IITP) grant funded by the Korea government (MSIT) (No. RS2020-II200882, SW STAR LAB, Development of deployable learning intelligence via self-sustainable and trustworthy machine learning), and the National Research Foundation of Korea (NRF) grant funded by the Korea government (MSIT) (No. RS-2024-00354036). Claudia Skok Gibbs is supported by an NSF-GRFP award (DGE-2234660). Kyunghyun Cho is the corresponding author.

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

## A  Proofs and Definitions

**Proposition 1.** *A set of marginal causes of* $x_i$ *is the subset of* $An(x_i)$.

*Proof.* Consider the ancestor structure of node $x_i$, defined as the subgraph of $\mathcal{G}$ containing the node set $x_i \cup An(x_i)$ and all corresponding edges. We denote this subgraph by $\mathcal{G}_i$. For each node $x_j \in x_i \cup An(x_i)$, we define the parent set restricted to this subgraph as $Pa(x_j \mid \mathcal{G}_i) = Pa(x_j) \cap An(x_i)$. Under the assumption of no latent variables in $\mathcal{G}$, the joint distribution over the ancestor structure factorizes as follows: $p(x_i \cup An(x_i)) = \prod_{x_j \in x_i \cup An(x_i)} p(x_j \mid Pa(x_j \mid \mathcal{G}_i))$. For any node $x_k \notin x_i \cup An(x_i)$, a do-operation on $x_k$ does not affect the ancestor structure since the node $x_k$ and all edges connected to $x_k$ are absent from $\mathcal{G}_i$. Also, there are no causal effects on the ancestor set $\mathcal{G}_i$ originating from nodes outside this set. Consequently, for all values $c$, we have $p(x_i \mid do(x_k = c)) = p(x_i)$. According to Definition 2, it follows that $x_k \notin x_i \cup An(x_i)$ is not a marginal cause of $x_i$. This completes the proof by contraposition. □

**Proposition 2.** *For any variable subset* $V \subset \mathcal{V}$ *s.t. containing* $x_i$ *and all root variables in* $\mathcal{G}$, *let* $An(x_i; V)$ *denote the set of ancestors of* $x_i$ *in the system consisting of* $V$, *following the marginalization rule of Bongers et al. (2016). Similarly, let* $Pa(x_i; V)$ *denote the set of parents of* $x_i$ *within* $V$. *Then,* $An(x_i; V) = An(x_i) \cap V$. *However, a counterexample exists for parents:* $Pa(x_i; V) \neq Pa(x_i) \cap V$.

*Proof.* Note that the marginalization rule of Bongers et al. (2016) on a node $x_l$ operates as follows: (1) every direct path $x_i \to x_l \to x_j$ is replaced with a direct edge $x_i \to x_j$, and (2) all remaining edges connected to $x_l$ are removed. This marginalization rule preserves all causal relationships (i.e., the existence of a directed path between nodes) after the removal of a node. Therefore, by definition, we have $An(x_i; V) = An(x_i) \cap V$. In contrast, for direct causes (parents), the relationship may not hold. Consider a causal chain graph with three variables: $x_1 \to x_2 \to x_3$. By definition, $Pa(x_3) = \{x_2\}$. However, in the subsampled system with $V = \{x_1, x_3\}$, $x_1$ becomes the direct cause of $x_3$, i.e., $Pa(x_3; V) = \{x_1\}$. On the other hand, $Pa(x_3) \cap V = \emptyset$. Thus, $Pa(x_i; V) \neq Pa(x_i) \cap V$. □

**Proposition 3** (Algorithm complexity). *Let* $n'$ *and* $m'$ *denote the subsampled variable and observation sizes, and let* $T$ *denote the ensemble size. For a dataset* $X$ *with* $n$ *variables and* $m$ *observations, the inference complexity of our algorithm is* $O(nm'T(n' + m'))$.

*Proof.* The set of $n$ variables can be partitioned into $\lceil \frac{n}{n'} \rceil$ inputs. For each input, the complexities of attention layers are $O(n'^2 m')$ and $O(n' m'^2)$, while the complexity of the feed-forward layers is $O(n'm')$. Thus processing each variable once results in a complexity of $O(\frac{n}{n'}(n'^2 m' + n' m'^2)) = O(nm'(n' + m'))$. Considering an ensemble size of $T$, the overall computational complexity becomes $O(nm'T(n' + m'))$. □

## B  Experimental Settings

### B.1  Model Architecture

For the feature extractor $g_\theta$, we utilize an axial-Transformer without positional encoding to ensure permutation equivariance (Ho et al., 2019), *i.e.*, if the variables are permuted, the model produces a correspondingly permuted causality prediction. Each Transformer layer comprises two attention layers, one along the variable dimension and one along the observation dimension, followed by a feed-forward layer (Figure 10). Both attention and feed-forward layers include layer normalization and a skip connection (Ba et al., 2016). Detailed configuration is provided in Table 3.

As described in Section 4.1, the input to the Transformer is a stack $[X, M] \in \mathbb{R}^{n \times m \times 2}$, and the output is two feature matrices $F_1, F_2 \in \mathbb{R}^{n \times d}$. In a basic axial Transformer, the output size for an input of size $n \times m \times 2$ is $n \times m \times d$, where $d$ is the embedding dimension (Figure 10). We denote this output as $H$. To obtain the feature matrix of size $n \times d$, we first average the output $H$ of size $n \times m \times d$ along the observation dimension $m$, resulting in a matrix of size $n \times d$. We then apply two feed-forward layers and concatenate the results to produce two feature matrices $F_1, F_2 \in \mathbb{R}^{n \times d}$. The total number of trainable parameters of our model is 62k. We will release the code for our model as open-source.

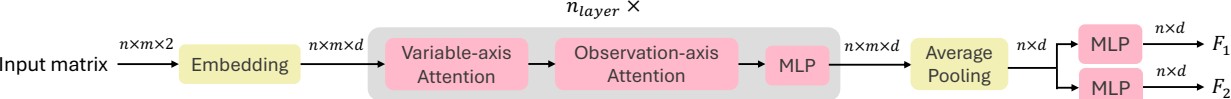

Figure 10: Computational flow of our feature extractor $g_\theta$. We indicate the size of the inputs for each module above the arrows, with only the embedding and pooling modules changing the sizes. For simplicity, normalization and skip connections are excluded from the figure. *MLP* indicates a two-layer perceptron with non-linear activation (Vaswani et al., 2017).

Table 3: Architecture configuration.

| Argument | Value |
| --- | ---: |
| Number of Transformer layers | 10 |
| Embedding dimension | 16 |
| Number of attention heads | 16 |
| Feed-forward layer hidden dimension | 96 |

### B.2    Training/Inference Configuration

**Training configuration.**    We train a neural network using the AdamW optimizer (Loshchilov & Hutter, 2019), with training configurations detailed in Table 4. The batch size is chosen to fully utilize our GPU memory (24GB). Training runs for a maximum of 40,000 steps, with early stopping applied if the validation accuracy does not improve over a span of 4,000 steps, checked every 200 steps. The limit of 40,000 training steps is set empirically, as we observed no further performance gains beyond this point on the training dataset specified in Table 6.

We tune the learning rate from the set {6e-4, 8e-4, 1e-3}, finding that 8e-4 yields the most stable training performance across all experimental settings. Notably, the optimal learning rate depends primarily on the model configuration rather than on the dataset type. To further enhance training stability, we apply a cosine learning rate scheduler, which reduces sensitivity to initial learning rates (Loshchilov & Hutter, 2017). Dropout is not used in our experiments, as it empirically decreases performance. We hypothesize that the subsampled training and inference scheme, in which predictions are made based on partial information, inherently reduces the need for dropout regularization.

**Inference configuration.**    Figure 9 describes the configurations required for our inference procedure (Algorithm 2). Hyperparameters are selected based on the analysis in Section 5.4. Note that we use the same sizes, $n'$ and $m'$, during training. The observation size is set to 200, identical to the variable size, to fit within our GPU memory constraints during training.

**Computing environment.**    We conduct all experiments including training and inference, using a NVIDIA RTX 3090 GPU with 24GB memory. The training time for models in Section 5.2 is approximately 9h, while inference takes about a few seconds per target (Table 8).

Table 4: Training configuration.

| Argument | Value |
| --- | ---: |
| Batch size | 32 |
| Training step | 40,000 |
| Learning rate | 8e-4 |
| Learning rate scheduler | cosine |
| Weight decay | 1e-5 |

Table 5: Inference configuration.

| Argument | Value |
| --- | ---: |
| Subsampled variable size $n'$ | 200 |
| Subsampled observation size $m'$ | 200 |
| Ensemble size $T$ | 50 |

### B.3 GRN Simulator

We use the SERGIO GRN simulator in our experiments (Dibaeinia & Sinha, 2020). Given a user-defined GRN, the simulator generates a gene expression matrix based on a specified cell type configuration.

**Generation mechanism.** The simulator samples gene expressions from the steady state of a dynamic system modeled as stochastic differential equations (Dibaeinia & Sinha, 2020). Master regulators (*i.e.*, root nodes in the causal graph) operate independently without external regulatory inputs, evolving with fixed production and decay rates. The regulatory influence of each gene is represented by a Hill function with pre-determined interaction parameters (Chu et al., 2009). This mechanism captures non-linear relationships and time-lagged effects, providing a realistic model of gene behavior.

**Technical noise.** The simulator produces datasets that reflect the statistical properties of real-world single-cell experimental data, incorporating several types of measurement errors and technical noise. The simulator applies these technical noises sequentially to the expression data sampled from the stochastic differential equations:

1. Dropouts: A high proportion of gene expressions (typically 60-95%) are randomly set to zero, simulating the dropout effect common in single-cell technologies.

2. Outlier genes: Some genes are assigned unusually high expression levels, replicating the presence of outliers.

3. Library size: The total expression level for each cell (known as library size) follows a log-normal distribution, reflecting the variability.

**Observational fidelity.** To generate the unique molecular identifier (UMI) count expression matrix, a quantification scheme in single-cell RNA-sequencing, the simulator applies Poisson random sampling to the expression values $\lambda$ after incorporating the technical noises (Chen et al., 2018). That is, the final observation value $v$ is derived as $v \sim \text{Poisson}(\lambda)$. To evaluate the impact of this Poisson sampling process on targeted cause discovery performance and to determine the performance ceiling of our method, we control the fidelity of the Poisson sampling and define three levels:

1. High fidelity: Uses expression $\lambda$ directly as the observation.

2. Medium fidelity: Uses the mean of 100 samples drawn from $\text{Poisson}(\lambda)$.

3. Low fidelity: Uses a single sample drawn from $\text{Poisson}(\lambda)$.

In Figure 4, we analyze the performance differences across varying levels of the simulator's observational fidelity. Unless otherwise specified, we use the high-fidelity setting for our analysis.

**Intervention.** We use the interventional setting identical to Lorch et al. (2022). Specifically, we perform gene knockout by setting the expression level of a specific gene to zero. We sample 10 intervened observations per gene. For example, given a GRN with 1000 genes, this results in an interventional dataset with 10,000 observations. From these observations and variables, we randomly sample subsets of size $n' = 200$ and $m' = 200$, as provided in Table 5.

### B.4 Simulated Dataset

We summarize the configurations of datasets used in our experiments in Table 6, adopted from Lorch et al. (2022). We generate training data using random graphs while testing on biological GRNs, E. coli (1,565 genes) and yeast (4,441 genes), obtained from Marbach et al. (2009). Figure 11 shows the degree histograms of these graph structures, highlighting the different patterns between biological graphs and random graphs. We generate interventional data by performing gene knockout on each gene, obtaining 10 observations per intervention. For yeast, we obtain 5 observations per intervention due to its larger gene count. We include 500 observational data points and conduct inference using a mixture of observational and interventional data. We preprocess each observation matrix using $\log_2$ counts-per-million (CPM) normalization following previous works (Lorch et al., 2022).

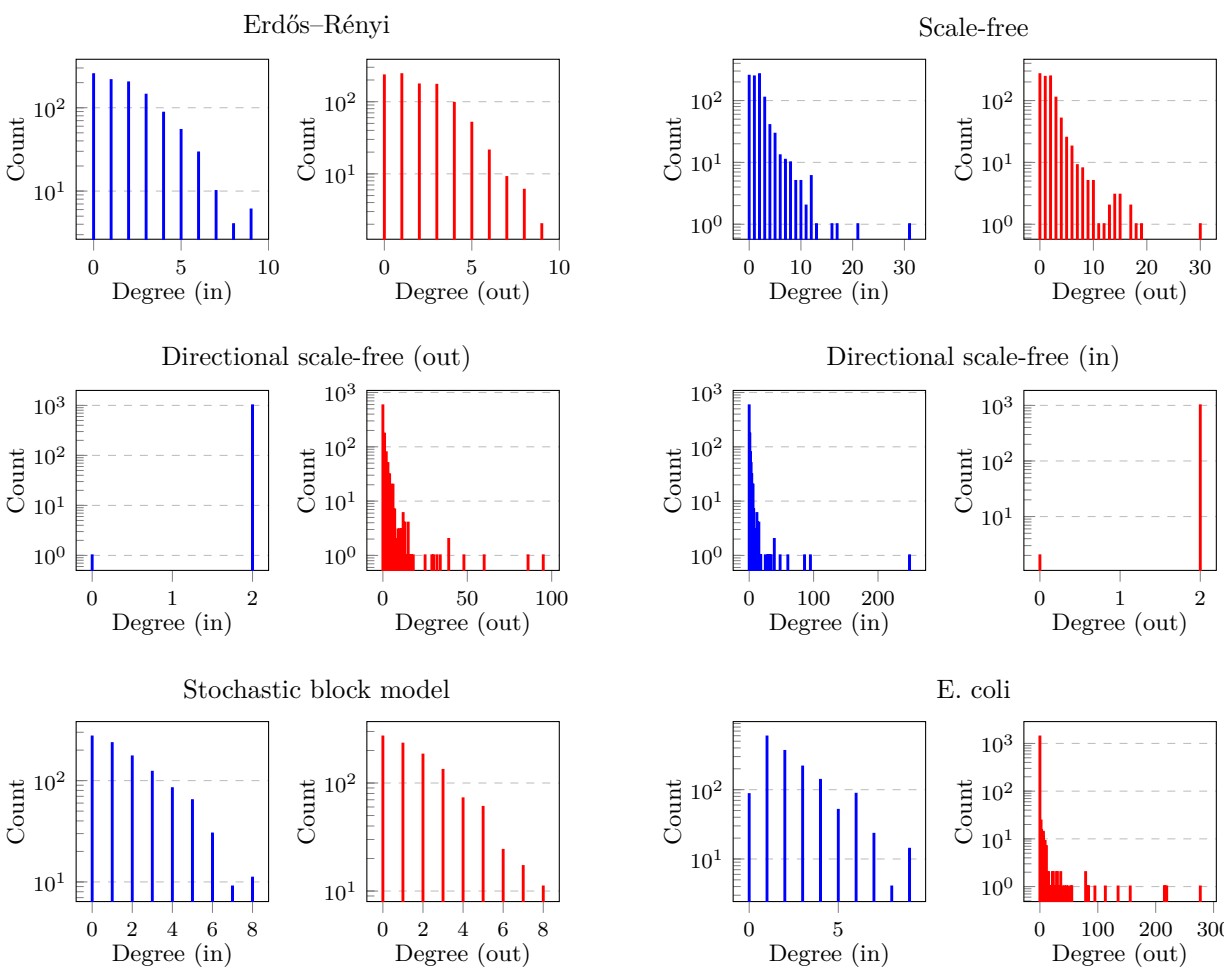

Figure 11: **Edge degree histograms** of graph structures considered in our experiments. All graphs have an average degree of 2. Blue represents the in-degree histogram, and red represents the out-degree histogram.

For the out-of-distribution (OOD) analysis in Figure 5, we adopt configurations from Lorch et al. (2022)-Table 4. These configurations include varying mechanism function parameters, such as Hill function coefficients and decay rates, which differ from the training settings. We also test models on OOD technical noise types, as described in Table 6. These noise types reflect the statistics of different experimental datasets, which have varying dropout percentages, outlier ratios, and library size distributions (Lorch et al., 2022).

**Computational challenge.** For the yeast dataset, each input comprises approximately 100 million entries—2,000 times larger than the number of pixels in an image from ImageNet (Deng et al., 2009) and 500,000 times larger than the number of input tokens for Vision Transformers (Dosovitskiy, 2021). This substantial size presents significant computational challenges for deep neural networks in memory usage and computation time, especially inducing out-of-memory issue in our computing environment described in Appendix B.2. To overcome this, we propose a subsampled-ensemble inference strategy that efficiently processes datasets across all considered scales.

### B.5 Human Cell Dataset

This section describes a Perturb-seq dataset used in the human cell experiments (Figure 6). The dataset contains gene expression data from the K562 cell line, which is derived from a patient with chronic myelogenous leukemia (Replogle et al., 2022). The dataset includes both interventional and observational data on

Table 6: **Dataset configuration.** Note for abbreviations used: ER (Erdős–Rényi), SF (Scale-Free), SF-direct (directional Scale-Free), and SBM (Stochastic Block Model) (Drobyshevskiy & Turdakov, 2019). For the training data, we randomly select the graph structure and edge degree independently from the candidate sets. We use a slash (/) symbol to separately denote the statistics for E. coli and yeast GRNs. For the exact configuration of technical noise, please refer to Table 4 in Lorch et al. (2022).

| Argument | Training set | Test set |
|---|---|---|
| Graph structure | {ER, SF, SF-direct, SBM} | E. coli/yeast |
| Average edge degree | {2,4,6} | 2.3/2.1 |
| Dataset size $|\mathcal{D}|$ per graph structure | 150 | 10 |
| Variable size ($n$) | 1,000 | 1,565/4,441 |
| Number of observations per intervention | 10 | 10/5 |
| Observation size (interventional) | 10,000 | 15,650/22,205 |
| Observation size (observational) | 500 | 500 |
| Number of cell types | 10 | 10 |
| Technical noise type | 10x-chromium | 10x-chromium |
| Technical noise type (OOD) | - | {illumina, drop-seq, smart-seq} |

Table 7: Predicted causal gene interactions for target gene MYC by our method. It is worth note that the absence of literature does not imply the invalidation of causality but rather indicates that the causality has not yet been elucidated.

| Target gene | Causal gene | Supported by |
|---|---|---|
| MYC | EEF1A1 | TCD-DL, STRING, Literature (Li et al., 2023; Wilson et al., 2024), Correlation |
| | NPM1 | TCD-DL, STRING, Literature (Hong et al., 2023) |
| | PTMA | TCD-DL, STRING, Literature (Lin et al., 2015) |
| | RPL26 | TCD-DL, STRING, Literature (Gong et al., 2023) |
| | RPL4 | TCD-DL, STRING, Literature (Egoh et al., 2010) |
| | RPS14 | TCD-DL, STRING, Literature (Zhou et al., 2013) |
| | RPS15A | TCD-DL, STRING, Literature (Liang et al., 2019) |
| | RPS6 | TCD-DL, STRING, Literature (Ravitz et al., 2007) |
| | RPL23A | TCD-DL, STRING |
| | NCL | TCD-DL, STRING |
| | HSPA8 | TCD-DL, STRING |
| | HIST1H2AE | TCD-DL, STRING |
| | RPL13 | TCD-DL, STRING |
| | ACTB | TCD-DL, STRING |
| | TUBB | TCD-DL, STRING |
| | TUBA1B | TCD-DL, STRING |
| | RPL26 | TCD-DL, STRING |
| | RPS18 | TCD-DL, STRING |
| | LST | TCD-DL |
| | VPS37C | TCD-DL |

gene expression. The intervention is performed through gene knockouts, identical to our simulation setting. From the Perturb-seq dataset, we obtain 1,868 genes that have undergone intervention. We conduct the inference among these genes. For each intervention, we randomly subsample 10 observations. We also sample 500 observational data points, consistent with the number used in our training setting (Table 6). Using this subsampled dataset, we run our TCD-DL inference algorithm to obtain cause scores for the target gene MYC, applying the same CPM normalization scheme used in our simulation data (Appendix B.3).

### B.6 Setting for Error Propagation Analysis

This section details the experimental setup for Figure 2 (Section 3). We generate scale-free random graphs with 100 nodes and define causal mechanisms using a two-layer perceptron with a Tanh activation function (Wu et al., 2024). Root variables are sampled from a uniform distribution. Both approaches—estimating the full causal structure and identifying a set of causal variables—are trained using the same Transformer architecture. However, their final layers are modified to align with their respective objectives. Both methods maintain the same number of model parameters and are trained on datasets of equal size. To enable a clear comparison of how the false negative rate (FNR) evolves with increasing cause-effect distance, we threshold the cause scores to ensure similar FNRs at a cause-effect distance of 1.

## C Additional Experimental Results

### C.1 Runtime Comparison

Table 8 compares the inference times of methods for targeted cause discovery. Some baseline methods (sortnregress and GENIE3) can individually compute a cause score vector for a target, while other baselines (AVICI, DCD-FG, PMF-GRN) require the calculation of the entire $n \times n$ score matrix to obtain a cause score vector for a target. From the table, our method conducts inference within seconds even for the yeast gene regulatory network (GRN) comprising 4,441 genes. In contrast, certain baselines encounter memory issues (AVICI) or require substantial computation time (DCD-FG, PMF-GRN).

Table 8: **Runtime measurement.** We measure inference time for identifying the causes of a target variable with an NVIDIA RTX 3090 GPU. *OOM* refers to the GPU out-of-memory error.

| Species | Sortnregress | GENIE3 | AVICI | DCD-FG | PMF-GRN | TCD-DL (ours) |
|---|---|---|---|---|---|---|
| E. coli (1,565 genes) | 0.9s | 1.7s | 3.1s | 1h 55m | 3h 2m | 2.5s |
| yeast (4,441 genes) | 2.8s | 2.7s | OOM | 98h 31m | 10h 46m | 7.8s |

### C.2 Small-Scale Settings

In Table 9, we conduct experiments in small-scale settings, comparing our approach to conventional methods such as PC and GIES (Spirtes et al., 2001; Hauser & Bühlmann, 2012). We train our model exclusively on synthetic datasets generated with randomly parameterized analytic functions (linear, non-linear multi-layer perceptron (MLP), polynomial, and sigmoid), while maintaining a scale-free random network structure with 10 nodes. For testing, we sample datasets with new causal mechanism coefficients and graph structures. Additionally, we evaluate the methods on a real-world dataset called Sachs, which contains 11 variables (Sachs et al., 2005). We obtain interventional data from bnlearn[1], including six intervention types, each applied to a single node. We sample a total of 100 observations for inference.

Given the DAG predicted by baseline methods, we apply topological sorting to obtain binary predictions for ancestor relationships. For our approach, we threshold the cause scores at 0 to obtain binary predictions. After inferring the causal structure, we measure binary classification accuracy for every pair of nodes to determine if the method accurately predicts causal (ancestor) relationships. Table 9 demonstrates that our method generalizes effectively to real-world settings with fewer nodes, highlighting its promising capabilities.

### C.3 Analysis on Human Cells

**Comparison to correlation ranking.** As illustrated in Figure 6, our method identifies novel causal factors for the gene MYC that are not captured by correlation ranking. To quantify the disparity between our model's predictions and correlation ranking, we analyze statistics across 1,868 genes from the Perturb-seq dataset. The average rank correlation between our model's cause scores and correlation-based rankings is

---

[1]https://www.bnlearn.com/book-crc/code/sachs.interventional.txt.gz

Table 9: **Performance on small graphs.** We measure binary classification accuracy on a pair of nodes to determine whether the method accurately predicts causality. TCD-DL (direct) refers to a variant of our method that trains a model to predict direct causal structures. Each column denotes test datasets with different causal structures and mechanisms. The number of nodes is 10, except for Sachs (Sachs et al., 2005), which has 11. For PC, we use only observational samples.

| Method | Linear | MLP | Polynomial | Sigmoid | Sachs |
|---|---|---|---|---|---|
| PC | 57.2 | 70.1 | 69.9 | 67.6 | 71.8 |
| GIES | 94.3 | 84.8 | 70.2 | 82.3 | 74.6 |
| TCD-DL (direct) | 96.5 | 86.4 | 82.3 | 89.7 | 75.5 |
| TCD-DL | **96.7** | **88.8** | **83.0** | **90.1** | **76.4** |

0.068, indicating low ranking similarity. When comparing the top 20 predictions from each approach, on average only 1.1 genes appear in both sets. Notably, an average of 6.4 genes from our top 20 predictions are validated by the STRING database, underscoring that our model identifies novel causal factors not captured by correlation.

**Identifying causes of leukemia-related genes.** To further validate our predictions from the K562 Perturb-seq dataset, we use the Human Protein Atlas to identify a set of 29 target genes associated with leukemia (Uhlén et al., 2015). We identify the top 10 causal predictions for each of these target genes, including those with support from the STRING database, and visualize these interactions in Figure 12. We present the resulting GRNs for each leukemia target gene and its predicted causal regulators through network diagrams in Figures 13 and 14. These visualizations highlight the potential regulatory roles of our identified causal genes, providing insights into the predicted interactions driving leukemia. Validation against STRING demonstrates that our approach generates well-supported and highly relevant causal predictions.

**Predicted influence of causal genes.** We investigate the regulatory influence of each predicted causal gene over leukemia-related target genes in Figure 15. Notably, nucleophosmin (NPM1) is predicted to regulate 25 of the 29 leukemia target genes, receiving support from the STRING database for 21 of these predictions. NPM1 mutations are prevalent in approximately one-third of adult Acute Myeloid Leukemia (AML) cases, leading to an abnormal cytoplasmic localization of the NPM1 protein (Falini et al., 2020). Although NPM1 mutations are primarily associated with AML, recent studies have identified them in a small subset of Chronic Myeloid Leukemia (CML) patients (Young et al., 2021). The prediction that NPM1 regulates a substantial number of leukemia-associated target genes is particularly significant as it provides insights into potential key regulatory mechanisms underlying leukemia pathology. Understanding how NPM1 influences these target genes in CML could reveal critical pathways involved in leukemia progression and help identify novel therapeutic targets.

## D   Extended Related Works

**GRN inference**   Gene regulatory network (GRN) inference aims to uncover directed causal influences among genes rather than merely identifying co-expression or correlation patterns (Badia-i Mompel et al., 2023). Representative approaches include ARCANE, which employs information-theoretic algorithms (Margolin et al., 2006), and GENIE3, which uses an ensemble of regression trees to predict each gene's expression based on all other genes (Huynh-Thu et al., 2010). While conventional causal discovery methods can be applied to GRN inference, they face significant challenges due to the large network sizes, complex causal mechanisms, and technical noise (Dibaeinia & Sinha, 2020). Recent advancements in sequencing technologies have enabled the collection of interventional data, such as gene knockout experiments, which provide more reliable identification of causal regulators (Replogle et al., 2022). Additionally, recent efforts have focused on developing simulators that generate single-cell gene expression data by modeling the stochastic nature of transcription based on user-provided causal graph structures (Dibaeinia & Sinha, 2020). Our data-driven learning approach leverages these advancements in data collection and simulation, demonstrating the potential of data-driven methods for GRN inference.

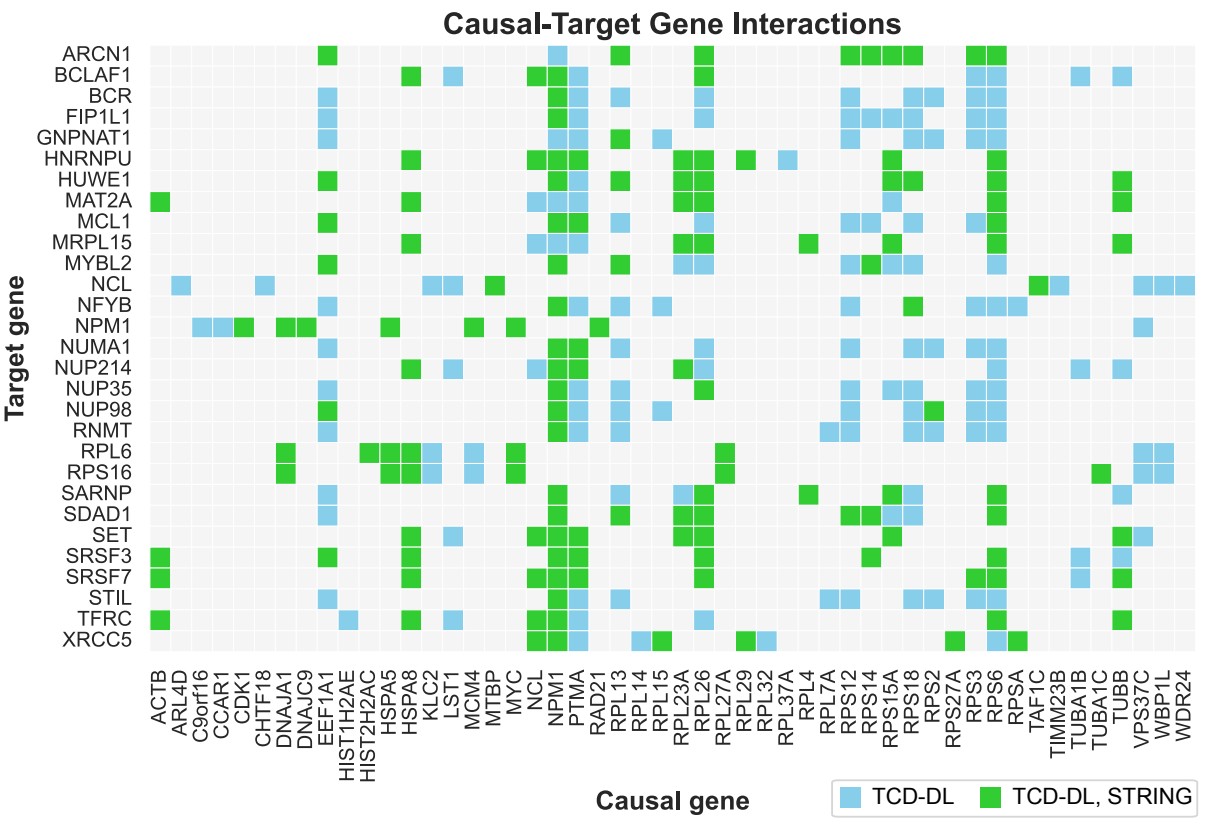

Figure 12: **Predicted causes of leukemia.** The matrix illustrates the predicted causality of leukemia-related target genes by TCD-DL predictions (blue) and TCD-DL predictions supported by STRING (green).

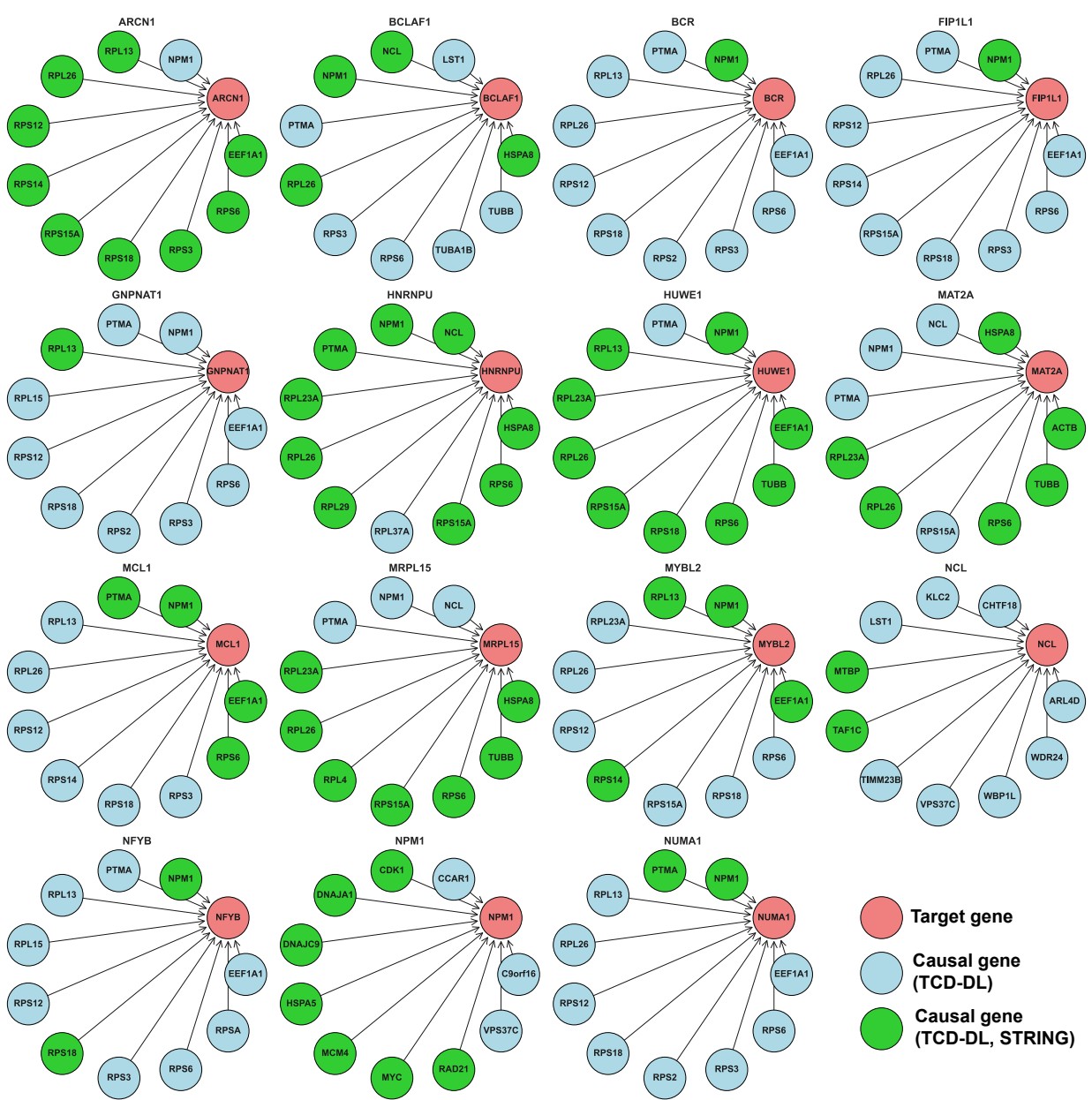

Figure 13: **Predicted causes of a leukemia-related gene.** GRNs illustrate the causal genes predicted by TCD-DL (blue) and TCD-DL predictions supported by STRING (green) for each leukemia-related target gene (pink).

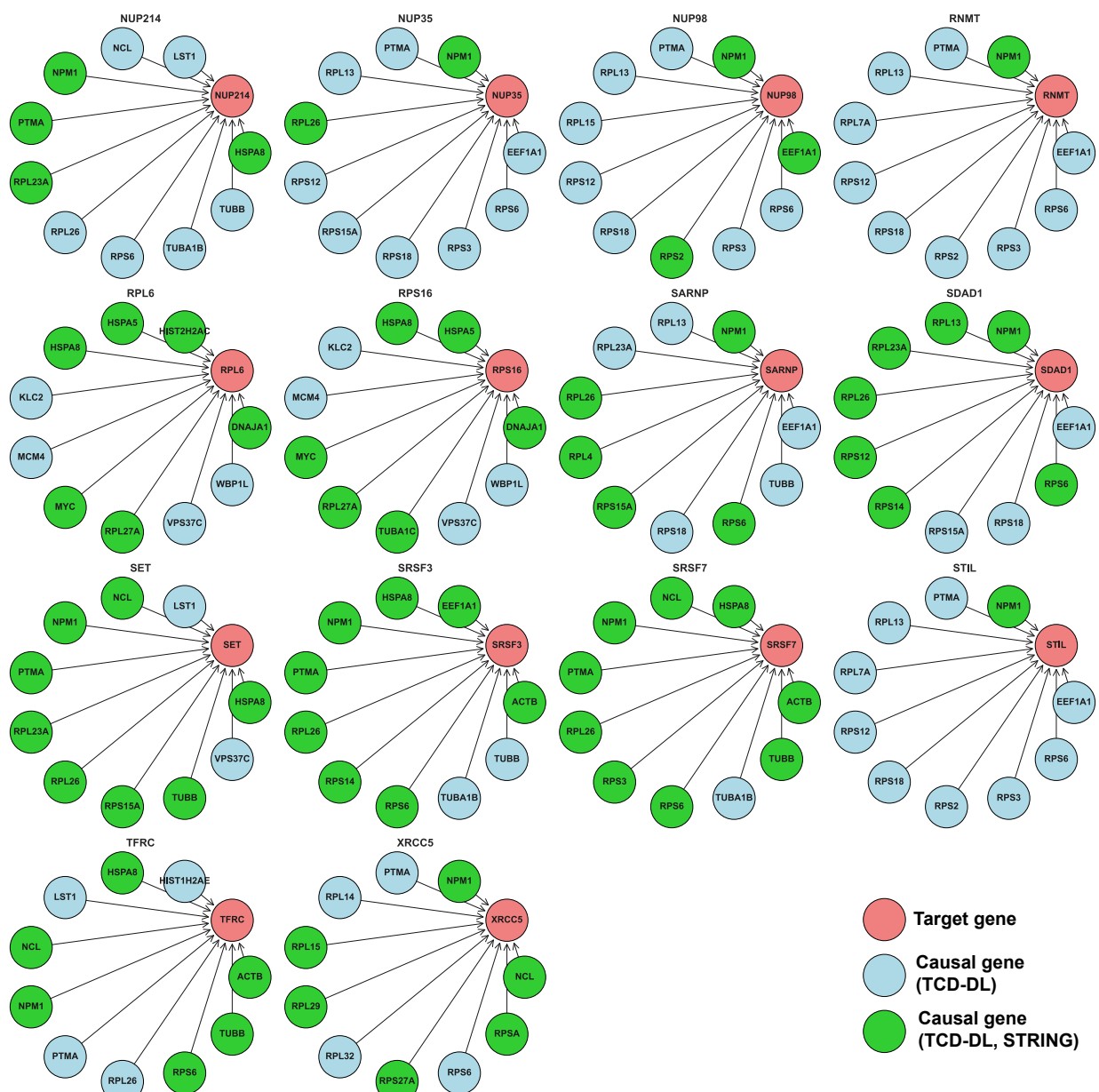

Figure 14: **Predicted causes of a leukemia-related gene.** GRNs illustrate the causal genes predicted by TCD-DL (blue) and TCD-DL predictions supported by STRING (green) for each leukemia-related target gene (pink).

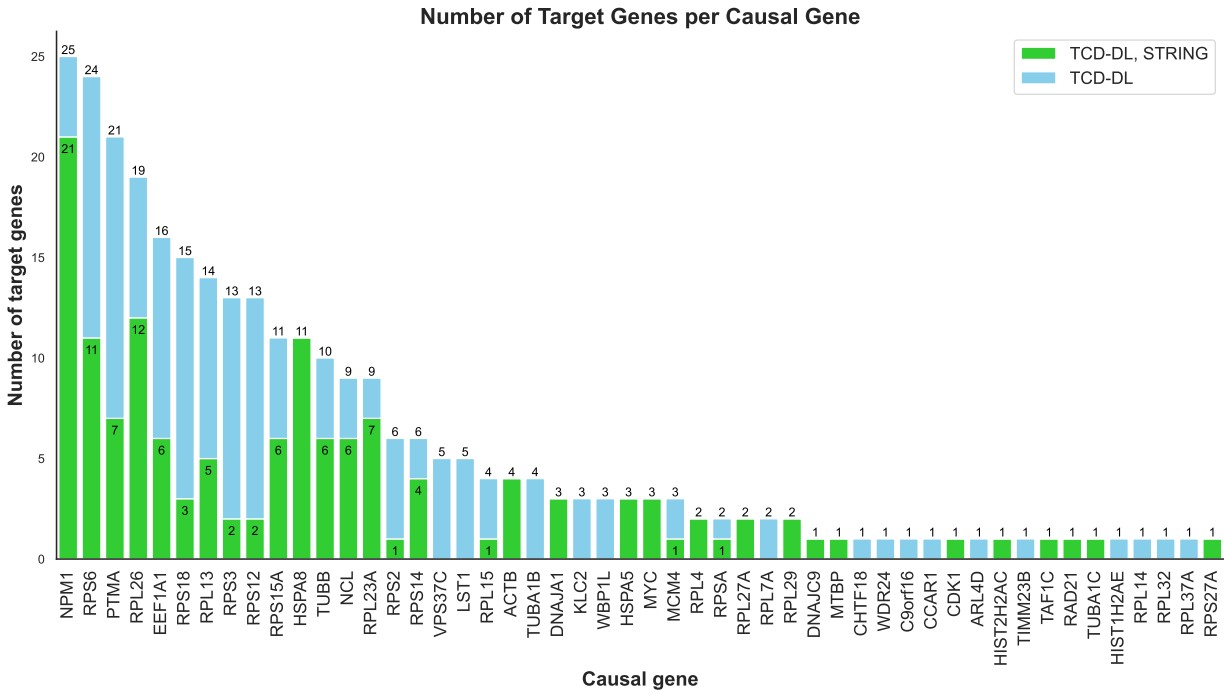

Figure 15: **Predicted influence of causal genes.** The histogram shows the number of leukemia-related target genes predicted to be regulated by each causal gene, with predictions made by TCD-DL (blue) and TCD-DL supported by STRING (green).

Table 10: **Benchmarking results.** Targeted cause discovery performance on E. coli GRN with 1565 genes over varying levels of simulator's observational fidelity. All measurements are expressed as percentages (%).

| Method | Fidelity high | | | Fidelity medium | | | Fidelity low | | |
|---|---|---|---|---|---|---|---|---|---|
| | AUROC | AP | F1 | AUROC | AP | F1 | AUROC | AP | F1 |
| Random | $50.0 \pm 0.0$ | $0.5 \pm 0.0$ | $0.5 \pm 0.0$ | $50.0 \pm 0.0$ | $0.5 \pm 0.0$ | $0.5 \pm 0.0$ | $50.0 \pm 0.0$ | $0.5 \pm 0.0$ | $0.5 \pm 0.0$ |
| Correlation | $51.1 \pm 8.0$ | $0.7 \pm 0.2$ | $0.8 \pm 0.1$ | $51.0 \pm 7.6$ | $0.7 \pm 0.2$ | $0.8 \pm 0.1$ | $49.8 \pm 3.3$ | $0.7 \pm 0.1$ | $0.8 \pm 0.1$ |
| Sortnregress | $49.8 \pm 0.7$ | $1.4 \pm 0.3$ | $1.3 \pm 0.4$ | $49.9 \pm 0.6$ | $1.4 \pm 0.4$ | $1.2 \pm 0.4$ | $50.1 \pm 0.1$ | $0.7 \pm 0.1$ | $0.7 \pm 0.2$ |
| PMF-GRN | $52.6 \pm 2.3$ | $0.9 \pm 0.1$ | $0.8 \pm 0.4$ | $52.9 \pm 3.2$ | $1.0 \pm 0.2$ | $1.0 \pm 0.3$ | $51.5 \pm 2.1$ | $1.0 \pm 0.1$ | $1.1 \pm 0.3$ |
| GENIE3 | $56.0 \pm 3.4$ | $2.9 \pm 0.8$ | $2.1 \pm 0.7$ | $54.3 \pm 3.9$ | $2.2 \pm 1.1$ | $1.5 \pm 0.9$ | $52.2 \pm 3.7$ | $1.1 \pm 0.2$ | $0.6 \pm 0.2$ |
| DCD-FG | $50.0 \pm 0.0$ | $0.6 \pm 0.0$ | $1.0 \pm 0.0$ | $50.0 \pm 0.0$ | $0.6 \pm 0.0$ | $1.0 \pm 0.0$ | $50.0 \pm 0.0$ | $0.5 \pm 0.1$ | $1.0 \pm 0.0$ |
| AVICI | $56.5 \pm 4.2$ | $1.0 \pm 0.2$ | $0.4 \pm 0.3$ | $62.7 \pm 5.5$ | $3.1 \pm 1.7$ | $3.7 \pm 2.6$ | $56.5 \pm 8.4$ | $1.2 \pm 0.9$ | $1.3 \pm 1.1$ |
| TCD-DL (ours) | $\mathbf{94.6} \pm 1.8$ | $\mathbf{38.6} \pm 6.3$ | $\mathbf{36.3} \pm 6.1$ | $\mathbf{81.7} \pm 11.3$ | $\mathbf{14.0} \pm 11.7$ | $\mathbf{13.4} \pm 12.0$ | $\mathbf{71.5} \pm 3.7$ | $\mathbf{2.4} \pm 1.5$ | $\mathbf{2.0} \pm 1.8$ |

