# OpenReview forum: "Large-Scale Targeted Cause Discovery via Learning from Simulated Data"
_TMLR — Accepted by TMLR_

### Review · Reviewer_gFu9 · 2025-06-04

**Summary Of Contributions:**

The paper considers the problem of cause discovery: finding all ancestors/causes of a variable in the causal graph. The authors detail arguments against using standard causal structure discovery to solve this problem, namely that causal structure discovery does not scale well with the number of variables, and that small errors in the predicted causal graph propagate, increasing the overall error rate for predicting ancestry. Instead, the authors propose Targeted Cause Discovery with Data-driven Learning (TCD-DL), which inputs a target variable as well as observational and interventional data with known intervention targets, and directly predicts for each other variable whether it is its ancestor or not, without reconstructing the causal graph.

TCD-DL is implemented by a Transformer architecture, and trained by stochastic gradient descent with a supervised learning objective using simulated data from synthetic graphs and known ancestry information as labels. The goal is to generalise the model to unseen causal structures and causal mechanisms. The main technical novelty of the TCD-DL architecture is its ability to scale linearly with the number of variables by utilising the *local inference strategy* proposed by the authors, which subsamples the input data both on the variable and the sample level.

The authors evaluate TCD-DL using causal structure discovery baselines. They show that it generalises better to unseen, realistic data after being trained on fully synthetic data even across different noise levels. The authors also validate their method on a real world human cell dataset. Finally, the authors provide extensive analysis and ablation studies for the impact of graphs and causal mechanisms seen during training, and other training and algorithm parameters.

**Audience:**

Yes

**Broader Impact Concerns:**

No concerns.

**Claims And Evidence:**

Yes

**Requested Changes:**

### Adjustments critical to securing recommendation for acceptance

In the example of the introduction, the authors claim that their approach “prioritizes transcription factors that are more accessible and cost-effective to manipulate”. However, as I understand, the proposed algorithm returns a binary label indicating whether a variable is an ancestor of some target variable or not, without outputting any further prioritization for cost-effectiveness. I suggest the authors describe how the algorithm provides such prioritization, or correct this claim otherwise.

### Adjustments that would strengthen the work

As described in the weaknesses, I believe the paper would benefit from providing more motivation for why “cause discovery” is an interesting and useful task.

The term “ensemble” and its parameter $T$ should be introduced in the following sentence: “Repeating this process across multiple subsamplings of variables and observations …”. Otherwise it is not introduced anywhere else before it is already used in the name of “ensembled local-inference”, as well as Proposition 3 and Algorithm 2.

In Section 5, the paper sometimes uses the phrase “targeted causal discovery” instead of “targeted cause discovery”. Since the former phrase was not introduced earlier, I assume this is a typo that should be corrected. Otherwise, please introduce targeted “causal” discovery and describe the difference compared to targeted “cause” discovery.

I find the usage of the term “local” in the paper confusing or overloaded. In Section 2.2, local approaches involve methods that discover the local neighborhood around a node, such as its Markov blanket or parent set. On the other hand, in Section 3 and later, local inference refers to the fact that subsampling the variables preserves ancestral relationships. I suggest the authors clarify how local inference in Section 3 and later relates to local methods in Section 2.2, or how it is “local” in general.

**Strengths And Weaknesses:**

### Strengths
I find the paper clearly written and easy to follow. The Method section describes the proposed algorithm in detail, with all design choices properly explained and motivated. The Experiment section is extensive, including relevant ablations and analysis. Overall, together with the hyperparameter details in the appendix, I feel that the work is complete and reproducible.

The proposed local inference strategy is a strong practical contribution of the paper, as it is simple and theoretically motivated by the Local Inference Guarantee in Proposition 2, while providing scaling benefits in the number of variables in practice as showcased by Section 5.2, which is a long-standing issue of causal structure learning.

### Weaknesses

It is not immediately clear how useful the task of cause discovery is compared to causal structure discovery, as the example given in the introduction mainly focuses on local causal discovery. While I agree that causal structure discovery is simply infeasible for systems with too many variables, I wish the authors provided more context and examples of use-cases that cause discovery, i.e. recovering only the set of ancestors, can solve. Are there other works in the literature that consider cause discovery, and if so, what is their motivation besides the task being more feasible?

---

> ### Author Response · Authors · 2025-08-06
> **Author rebuttal**
>
> Dear reviewer, we truely appreciate your time and efforts to provide valuable feedback for our work. We reflected all of the comments and **uploaded the revised manuscript**, coloring the major changes in red. In below, we leave detailed response to each of the feedback provided.
>
> ---
>
> **Q) more motivation for why “cause discovery” is an interesting and useful task.**
>
> - Thank you for the suggestion. We added a new paragraph introducing context of the cause discovery in the revision:
> -  In large-scale systems, inferring the complete set of causal relationships is often unnecessary, as various downstream tasks directly benefit from identifying only the causes that are relevant to specific outcomes [1]. For instance, in drug discovery, it is sufficient to identify causal transcription factors associated with disease-related genes, rather than mapping the entire gene regulatory network (GRN), to effectively guide therapeutic interventions [2]. Similarly, in epidemiology or policy evaluation, interventions can be effectively designed by pinpointing only the key causal drivers of infection rates or targeted outcomes, without modeling the full complexity of inter-variable dependencies [3]. These examples illustrate that discovering the causes of specific targets has practical utility while reducing the complexity inherent in large-scale causal discovery [1].
>
>   [1] Magliacane et al., “Ancestral Causal Inference”, 2016
>   [2] Hyunh-Thu et al., “Inferring regulatory networks from expression data using tree-based methods”, 2010.
>   [3] Glass et al., “Causal Inference in Public Health”, 2013
>
> ---
>
> **Q) I suggest the authors describe how the algorithm provides such prioritization, or correct this claim otherwise.**
> - Thank you for the pointer. As you suggested, we have toned down and corrected the claim by removing the statement regarding prioritization from the introduction. Instead, we have added example use cases of cause discovery, highlighting its practical utility as above.
>
> ---
>
> **Q) The term “ensemble” and its parameter $T$ should be introduced in the following sentence: “Repeating this process across multiple subsamplings of variables and observations …”.**
> - Thank you for your suggestion. We reflected your comment in our revision.
>
> ---
>
> **Q) I find the usage of the term “local” in the paper confusing or overloaded.**
> - Thank you for the pointer. We have replaced the term ‘local inference’ with ‘subsampled-ensemble inference’ and updated other related terms accordingly to enhance clarity.
> ---
>
> **Q) In Section 5, the paper sometimes uses the phrase “targeted causal discovery” instead of “targeted cause discovery”.**
> - Thank you for the pointer. We have updated all the phrases to "targeted cause discovery."
>
> ---
>
> We truly appreciate your constructive feedback. If you have any remaining concerns, please let us know.

---

### Review · Reviewer_JL6c · 2025-06-16

**Summary Of Contributions:**

This paper continues a line of work using what the authors call
'learning-based approaches' to causal learning. The distinguising
characteristic of this paper is that the goal is to identify causes of
a particular target variable and not to infer an entire causal
structure, or even to distinguish between direct and indirect causes.
Given this restricted goal the learning method can be tailored to it,
and so it is no surprise that the authors' method does better at
identifying causes than other methods aiming to infer causal structure
more generally.

The rationale for 'learning-based approaches' is better given by Lorch
et al than in the current paper, so I would encourage the authors to
spend a little time improving this aspect of the paper. Leaving
interventions aside, the method of generating simulated data (See B.4)
defines a joint distribution: P(G)P(Theta|G)P(x|G,Theta) where G is
the graph, Theta the "mechanism parameters (mech.), and noise
configurations" and x is an observation sampled from the distribution
defined by (G,Theta). The terms in the joint distribution
factorisation are structure prior, parameter prior and likelihood
respectively.

This way of describing the problem is essentially given by Lorch et al
and would encourage the authors to also include such a description
since it it's very helpful in allowing a reader to grasp the
essentials of the presented method.

If G(l_i[j]) indicates that xj is an ancestor of xi in G, then the
authors' goal is to estimate s_i[j](X) = Sum_{G:
G(l_i[j]=1)}P(G|X). Rather than attempt to do this by, say, sampling
graphs from the posterior P(G|X) using MCMC, the authors take a more
direct approach: use a transformer to learn (all the) s_i[j](X) from
simulated data. So I would argue that what we have here is a
particular way of doing Bayesian statistics, so it's success depends
on the choice of prior (how the random graphs are generated, how the
'mechanisms' are chosen). The method is somewhat similar to
"Approximate Bayesian computation".

The goal of identifying causes of a given target is sufficiently
important that I think a method for doing just this is justified.
Although the description of the method could be improved (see above) I
could work out what they were doing (although much was contained in
the appendices). The authors' method has high computational
complexity, but this issue has been addressed properly and a
reasonable subsampling method used.

The experimental results support the usefulness of the method.
Comparative results show the authors' method in a good light, although
one might view the comparison as a little unfair, given that the other
methods were not specifically designed for the task at hand.

Fig 8d shows that we really need interventions to do a good job of
getting the causal direction correct, but 68% on purely observational
data seems reasonable.

**Audience:**

Yes

**Claims And Evidence:**

Yes

**Requested Changes:**

Address the weakness mentioned above by:

1) improving the motivation/rationale for the 'learning-based approach'
2) Tone down the claims mentioned in (1) and (3) above
3) Fix small things like typos.

**Strengths And Weaknesses:**

The strengths of the paper are:

1) a reasonably successful method for causal identification is presented and evaluated.
2) the writing is reasonably clear
3) there are quite a few useful and interesting comparative experimental results presented
4) due attention is paid to computation issues

The main weakness is that the rationale for the 'learning based approach' should be fleshed out a little more. Here are some specific points:

(1) Fig 2 compares results of a particular 'Structure' method versus a
particular 'Cause' (ie the authors') method. They conclude from this
that: "The results show that our method maintains a stable error rate
regardless of distance, whereas error rates in causal structure
discovery methods increase as the cause-effect distance grows." This
is an invalid inference from results using a particular causal
structure discovery method to "causal structure discovery methods" in
general. This is wrong, of course.

(2) Why is the (reasonable) subsampling approach mentioned in S4.2 called
'local inference'?

(3) So we take top 20 genes found by the method. Unsurprisingly, most of
them have been found to be associated to the target. It is claimed
that 30% are validated by the literature (the 7 green genes in Fig 6)
but in only 5 cases is the causality the correct way round.

TYPOS:

A set of marginal causes of xi is the subset of An(xi ).
->
The set of marginal causes of xi is a subset of An(xi ).

Extened
->
Extended

---

> ### Author Response · Authors · 2025-08-06
> **Author rebuttal**
>
> Dear reviewer, we truely appreciate your time and efforts to provide valuable feedback for our work. We reflected all of the comments and **uploaded the revised manuscript**, coloring the major changes in red. In below, we leave detailed response to each of the feedback provided.
>
> ---
>
> **Q) Improving the motivation/rationale for the 'learning-based approach'**
> - Thank you for the suggestion. We have **added Subsection 2.2** to the revised manuscript, describing the motivation and rationale behind learning-based approaches.
> - Specifically, we explain the key interpretation of amortized variational inference provided by Lorch et al. for causal discovery, positioning our method within their interpretation while highlighting the key difference of our approach that performs direct variational inference on the ancestor set distribution.
>
> ---
>
> **Q) weakness1 on Figure 2. invalid inference from results using a particular causal structure discovery method to "causal structure discovery methods" in general.**
> - Thank you for the pointer. We have incorporated your comment into our manuscript and moderated the statement in Section 3, clarifying that the claim is based on **a controlled experiment using a specific causal discovery method**, while ensuring a fair comparison.
> - Revised text: "We compare a method estimating the full causal structure with a method identifying only a set of causes. To ensure a fair comparison, we use the same model architecture and dataset for both methods, **differing only in training objectives and inference strategies**. The results indicate that the error rate of a method identifying causes remains stable regardless of distance, whereas the error rate of a method trained to estimate the causal structure increases as the cause-effect distance grows."
>
> ---
>
> **Q)  It is claimed that 30% are validated by the literature (the 7 green genes in Fig 6) but in only 5 cases is the causality the correct way round.**
> - In Figure 6, a **yellow gene** is also validated by the literature in addition to the green genes. (Yellow represents predictions by correlation ranking). As you mentioned, 5 green genes have the correct causal direction, and including the yellow case, a total of 6 out of 20 (30%) cases are validated by the literature. We have clarified this information in our revision.
>
> ---
>
> **Q) Why is the (reasonable) subsampling approach mentioned in S4.2 called 'local inference'?**
> - Thank you for the pointer. We have replaced the term ‘local inference’ with ‘subsampled-ensemble inference’ and updated other related terms accordingly to enhance clarity.
>
> ---
>
> **Q) Fix small things like typos.**
> - Thank you for the pointer. We fixed all the typos mentioned.
>
> ---
>
> We truly appreciate your constructive feedback. If you have any remaining concerns, please let us know.

---

### Review · Reviewer_nsJv · 2025-07-31

**Summary Of Contributions:**

### Summary

This paper provides a method for predicting an ancestral set of a targeted node from a mixture of observational and interventinoal datasets. The proposed framework (TCD-DL) takes a  simulated training dataset, which is a triplet of an observational sample, an interventional sample and a graph. In the training phase, the model for computing the score vector, which indicates which nodes are ancestors of the target node, is trained from the random subsets of the training dataset with random subset of nodes with marginalized causal graphs.  In the test phase, the model can predict the ancestors of a given target node.

### Contribution

This paper recasts the causal discovery task into the classification task using so-called “Sim2Real” or “synthetic-data pretraining method”. This approach is interesteing, practically strong and scalable when the data generating process for training and test data are believed to be shared. Under this setting, the superior performance is shown from many experiments in the paper.

**Audience:**

Yes

**Claims And Evidence:**

Yes

**Requested Changes:**

### Issues: Overclaiming of Title and Contribution

I think the title and the contribution is overly claimed, because the method is confined to those which has the high-fidelity simulators. This is a very special type of causal discovery setting, which should be highlightened in the introduction and the title. This is not the same as a standard causal discovery problem. This kind of confinement should be reflected in Introduction and the title.

### Questions: Performance Trade-off of Local Subsampling

What is the performance loss when comparing the use of local subsampling versus not using it? I understand that local subsampling has been implemented to achieve scalability, but to what extent does it trade off with performance?

**Strengths And Weaknesses:**

### Strength

- The paper recasts targeted causal discovery as binary classification of ancestors (order-free) and combines it with local subsampling + ensembling, yielding practical scalability on large GRNs.
- The paper includes comprehensive experiments with real-world datasets and demonstrates significant practical impact.

### Weakness

1. **Heavy reliance on high-fidelity simluators**

    The primary limitation of this approach is its fundamental reliance on the quality and realism of the simulator used for training. The model's outstanding performance is contingent on the SERGIO simulator's ability to generate synthetic data that accurately captures the complex dynamics of real-world gene regulatory networks.

    This raises a critical question about the method's broader applicability: its success in new domains (e.g., economics, climate science) would depend entirely on the existence of an equally high-fidelity simulator for that specific domain. Therefore, while the method itself is powerful, its generalizability is ultimately bounded by our ability to accurately simulate the target system, a significant challenge in many scientific fields.

2. **Strong assumption in Proposition 2**

    The assumptions in Proposition 2 appear quite strong. To apply this result to the local subsampling, it seems that every local subsample would need to include the root set. Is this actually the case? If so, how does the method guarantee this condition?

---

> ### Author Response · Authors · 2025-08-06
> **Author rebuttal**
>
> Dear reviewer, we truely appreciate your time and efforts to provide valuable feedback for our work. We reflected all of the comments and **uploaded the revised manuscript**, coloring the major changes in red. In below, we leave detailed response to each of the feedback provided.
>
> ---
>
> **Q) Overclaiming of Title and Contribution (Heavy reliance on high-fidelity simluators)**
> - Following your suggestion, we have updated the title to “Large-Scale Targeted Cause Discovery via Learning from Simulated Data” in the revised manuscript, highlighting the use of simulated data in our method.
> - We also refined our proposal and experimental claims in the introduction to clearly convey our contributions. For example:
>   - “In this study, we propose a scalable method for identifying the causal variables … by leveraging data simulators.”
>   - “Empirical evaluations demonstrate that our method effectively identifies causal relationships … assuming the availability of high-fidelity simulators.”
>
> ---
> **Q) Strong assumption in Proposition 2**
> - Thank you for your pointer. We introduce Proposition 2 to compare the properties of ancestors and parents using the causal subgraph derived through marginalization rule by Bongers et al. (2016). As you pointed out, the assumption to include all root variables in the subsampling is strong and may not hold in practical scenarios.
> - To address this, in practice, we **generalize the results** from Proposition 2, defining $An(x_i; V) = An(x_i) \cap V$ for arbitrary subsets of variables. Thus, we apply random variable subsampling during both training and inference phases, as described in Algorithms 1 and 2 of Section 4.2.
>   - Given a subsampled variable set $V$, we define the label (a binary indicator of ancestors) for a target variable $x_i$, denoted as $l_{i,V} \in$ {$0,1$} $^{|V|}$. Each entry of the label is defined as 1 if the corresponding variable in $V$ belongs to $An(x_i) \cap V$, and 0 otherwise.
> - This procedure naturally extends the results from Bongers et al. (2016) and **ensures consistent labeling**. To illustrate, consider a variable $x_j \in An(x_i)$. For any subset $V$ containing both $x_i$ and $x_j$, we have $x_j \in An(x_i) \cap V$. Consequently, the variable $x_j$ receives a label of 1 within $l_{i,V}$ regardless of $V$, guaranteeing that our method provides consistent and accurate supervision signals throughout subsampled training.
> - We have explicitly incorporated this clarification into the revised manuscript’s method description in Section 4.2.
>
> ---
> **Q) Performance Trade-off of Local Subsampling**
> - To address your questions, we conducted experiments using the Sergio gene simulator to generate a synthetic causal graph and interventional datasets containing 200 nodes. The test dataset adopts a scale-free random graph structure that differs from the graphs used during training. We evaluated the model trained in the main experiment under various subsampling sizes:
> |Subsampled variable size|AUROC (%)|
> |--|--|
> |50|89.8|
> |75|92.8|
> |100|93.9|
> |150|95.5|
> |200 (no subsampling)|95.7|
>
>   *Table. Performance of our targeted cause discovery method on Sergio simulated data with 200 variables.*
>
> - The results indicate a concave decline in performance as subsample size decreases. Notably, our method still achieves **meaningful inference performance** even with smaller subsamples, such as 50 variables.
> - For large-scale datasets like E. coli (1,565 genes) and yeast (4,441 genes) used in our main experiments, processing the entire dataset simultaneously with a Transformer architecture is **computationally infeasible** due to significant memory demands, leading to Out-of-Memory (OOM) issues under our computing environment of 8 × 24GB NVIDIA RTX3090 GPUs.
>   - As detailed in Appendix B.4, the yeast's observation matrix comprises approximately **100 million entries**, comparable to thousands of high-resolution images in ImageNet [1]. (Typically, deep learning approaches employ mini-batch stochastic gradient descent rather than full-batch training to manage such computational constraints [2].) Accordingly, local subsampling is crucial for handling large-scale observational data effectively.
> - Figure 9-b presents results on the E. coli dataset by varying the subsample size up to 300 variables, avoiding OOM issues. Interestingly, increasing subsample size from 200 to 300 variables resulted in decreased performance. This suggests that when input complexity exceeds a certain threshold, inference performance can deteriorate, highlighting a trade-off between information sufficiency and complexity of subsampled dataset. This indicates a local inference strategy can **outperform full dataset inference** in certain cases.
>
> [1] Deng et al., Imagenet: A large-scale hierarchical image database. CVPR, 2009.
> [2] Goodfellow et al., “Deep Learning”, MIT Press, 2015
>
> ---
> We truly appreciate your constructive feedback. If you have any remaining concerns, please let us know.

---

### Decision · Action_Editor_LJWY · 2025-09-21

**Recommendation:** Accept as is

**Additional Comments:**

Reviewers are quite satisfied with the changes from the discussion period. One reviewer suggests, additionally, in the final version, to add a brief discussion / recommendation on how to proceed when high-fidelity simulators are not available. (perhaps through a pointer to other methods applicable in that case).

**Audience:**

Yes

**Audience Explanation:**

In agreement with the reviewers, I can assert that the topic of cause discovery is of high importance in the TMLR community. The motivation in the introduction has been strengthened after review, and the experiments include an important real-world example application to genetics.

**Claims And Evidence:**

Yes

**Claims Explanation:**

I agree with the reviewers' unanimous decision that indeed the claims are well supported. The work claims the proposed method is scalable and effective at directly inferring the set of causes of a target, within an unknown causal graph, given certain simulated data. From the review process, the conditions for the claims have been tightened, emphasizing the importance of an high-fidelity simulation. In the form presented in the the latest revision, the claims are convincing and well justified by the results.